# Complexity of frequency receptive fields predicts tonotopic variability across species

Quentin Gaucher[†], Mariangela Panniello[†], Aleksandar Z Ivanov, Johannes C Dahmen, Andrew J King, Kerry MM Walker*

Department of Physiology, Anatomy & Genetics, University of Oxford, Oxford, United Kingdom

**Abstract** Primary cortical areas contain maps of sensory features, including sound frequency in primary auditory cortex (A1). Two-photon calcium imaging in mice has confirmed the presence of these global tonotopic maps, while uncovering an unexpected local variability in the stimulus preferences of individual neurons in A1 and other primary regions. Here we show that local heterogeneity of frequency preferences is not unique to rodents. Using two-photon calcium imaging in layers 2/3, we found that local variance in frequency preferences is equivalent in ferrets and mice. Neurons with multipeaked frequency tuning are less spatially organized than those tuned to a single frequency in both species. Furthermore, we show that microelectrode recordings may describe a smoother tonotopic arrangement due to a sampling bias towards neurons with simple frequency tuning. These results help explain previous inconsistencies in cortical topography across species and recording techniques.

## Introduction

The primary sensory areas of the cerebral cortex are characterized by a systematic spatial arrangement of responses to the stimulus features represented by peripheral receptors. In A1, this takes the form of a logarithmic gradient of neural response preferences for a single 'best' frequency, known as tonotopy (*Winer and Lee, 2007*). Early electrophysiological investigations revealed a tonotopic arrangement of neuronal frequency preferences in A1 in all species studied, including monkeys (*Merzenich and Brugge, 1973*), cats (*Hind, 1953*), ferrets (*Kelly et al., 1986a*), gerbils (*Steffen et al., 1988*), and rats (*Sally and Kelly, 1988*). More recent multielectrode recordings (*Imaizumi et al., 2004*; *Bizley et al., 2005*; *Recanzone et al., 1999*) and large-scale imaging experiments (*Nelken et al., 2008*) have confirmed the presence of a tonotopic gradient over large areas of A1 in carnivores and primates, while single-neuron recordings in awake animals have reported clear frequency gradients in marmosets (*Bendor and Wang, 2008*) and weaker tonotopy in cats (*Goldstein et al., 1970*).

In vivo two-photon calcium imaging can measure the stimulus preferences of a much larger number of individual neurons within a cortical region than the above techniques, improving the spatial sampling of mapping studies. The results of such experiments in mouse A1 over the past decade have challenged our understanding of the cortical tonotopic map. While previous electrophysiological and large-scale imaging studies confirmed an overall tonotopic organization in mouse A1 (*Hackett et al., 2011*; *Guo et al., 2012*; *Stiebler et al., 1997*), more recent two-photon imaging studies have reported different degrees of frequency tuning variability among neighboring neurons, from well-ordered local maps (*Issa et al., 2014*) to moderately heterogeneous tuning (*Bandyopadhyay et al., 2010*; *Rothschild et al., 2010*; *Panniello et al., 2018*; *Romero et al., 2020*). Similarly, two-photon imaging has revealed local heterogeneity in whisker selectivity in mouse

*For correspondence:
kerry.walker@dpag.ox.ac.uk

[†]These authors contributed
equally to this work

Competing interest: See
page 21

Reviewing editor: Brice
Bathellier, CNRS, France

primary somatosensory cortex (*Kerr et al., 2007*; *Sato et al., 2007*), and in orientation tuning in mouse primary visual cortex (V1) (*Bonin et al., 2011*).

There are several possible explanations for the local variation of A1 frequency preferences revealed by two-photon imaging studies compared to the smoother tonotopy described in electrophysiological studies. First, the heterogeneity may be due to the higher spatial sampling rate of two-photon imaging. Microelectrode studies are often based on responses that are summed across several neurons, and this spatial averaging may lead to smoother tonotopic gradients (*Guo et al., 2012*). Even in single neuron recordings, the spacing between cells is at least 50 μm (*South and Weinberger, 1995*), and typically ~100 μm or greater, so variations in stimulus preferences between neighboring cells are not usually examined (*Kanold et al., 2014*). Secondly, multielectrode recordings may be biased more towards the most robustly responding neurons in thalamorecipient layers, whereas most two-photon imaging studies have been restricted to the superficial layers of the cortex. Studies investigating layers 2/3 and 4 have usually found smoother tonotopy in the deeper layers (*Guo et al., 2012*; *Winkowski and Kanold, 2013*) (but see *Tischbirek et al., 2019*, reporting similar tonotopic organization across all layers of mouse auditory cortex). We tested these two explanations by comparing the frequency tuning of neurons in superficial auditory cortex measured using in vivo two-photon calcium imaging and high-density multielectrode (Neuropixels; *Jun et al., 2017*) recordings.

The variations in tonotopy across studies may also partly reflect a species difference in the cortical organization of rodents and higher mammals, particularly as local thalamic inputs to A1 in mice are also heterogeneous in their frequency tuning (*Vasquez-Lopez et al., 2017*). Two-photon imaging studies of primary visual cortex have revealed poorer spatial organization of orientation tuning in rodents than in cats (*Bonin et al., 2011*; *Ohki et al., 2005*), tree shrews (*Lee et al., 2016*), and ferrets (*Wilson et al., 2017*). The same may be true of tonotopy in A1. This view is supported by a recent study in marmosets, in which A1 was reported to be more tonotopically organized than in rats (*Zeng et al., 2019*). The present experiments further examined whether local heterogeneity in tonotopy is a general feature of mammalian A1, or a peculiarity of rodents. We conducted two-photon imaging experiments in mice and ferrets, and compared their local tonotopic organization.

It is not clear how spatial organization of tuning to a single sound frequency coincides with the known functions and complex frequency receptive fields of auditory cortex. Throughout the ascending auditory pathway, neurons progressively integrate spectral and temporal features of sound (*Linden and Schreiner, 2003*), and the receptive fields of many A1 neurons are poorly predicted by a model of linear tuning to a single sound frequency (*Ahrens et al., 2008*; *Sadagopan and Wang, 2009*). Recent studies have shown that the preferred frequencies of A1 neurons with irregular tuning curves are poorly mapped (*Romero et al., 2020*; *Tischbirek et al., 2019*). Our present results further demonstrate how A1 neurons with multipeaked frequency tuning curves impact on tonotopic organization.

This is the first application of 2-photon calcium imaging and Neuropixels recordings to study auditory processing in ferrets. Ferrets are a popular animal model in auditory neuroscience because they have a gyrencephalic cortex, their A1 is easily accessible, they are readily trained on behavioural tasks, and (unlike mice) their hearing range overlaps well with the human audiogram (*Nodal and King, 2014*).

We found that mice and ferrets show comparable local variation in frequency preferences among well-tuned neurons. Furthermore, neurons with more complex frequency receptive fields have poorer tonotopic arrangement in both species. Our findings suggest that ferrets and mice may use the same general principles for mapping frequency in A1, and explain past discrepancies in cortical maps described in studies using different species and experimental techniques.

## Results

We investigated how the frequency preferences of neurons in the superficial layers of ferret A1 are organized spatially, both locally (within ~0.25 mm$^2$) and along the entire tonotopic axis (~3.5 mm), and how this organization compares to the local tonotopy in mice (*Bandyopadhyay et al., 2010*; *Rothschild et al., 2010*; *Panniello et al., 2018*). We used an AAV vector to express either GCaMP6m (four subjects) or GCaMP6f (four subjects) in the right A1 of ferrets (*Figure 1A*; *Figure 1—figure supplement 1A,B*) and recorded neuronal calcium transients in 3604 neurons across

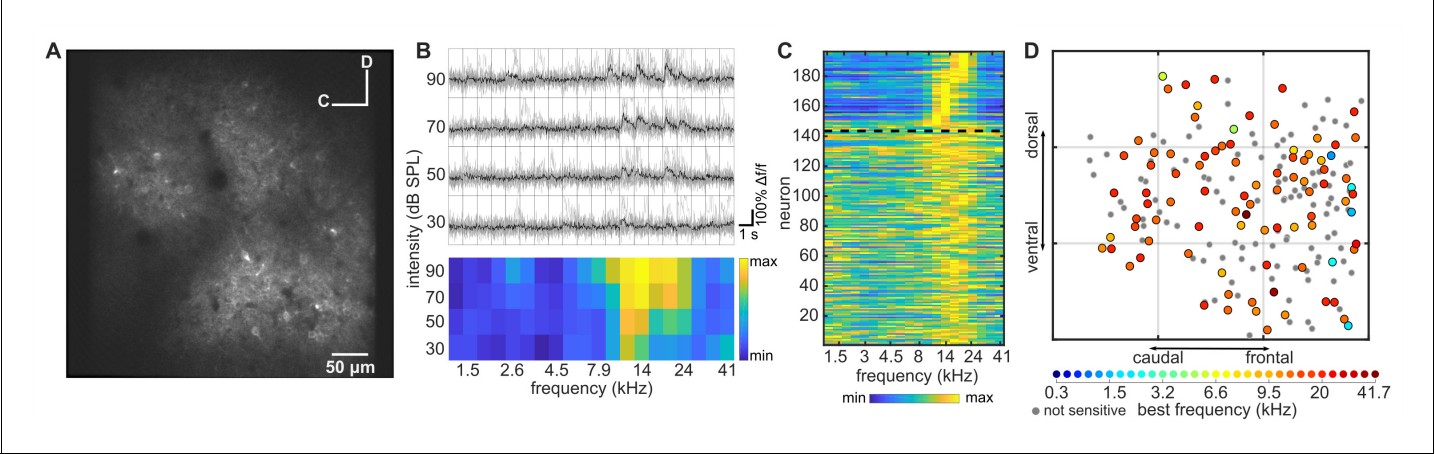

**Figure 1.** Imaging neuronal responses to pure tones in ferret A1. (A) An example cortical field in A1 of ferret 1, imaged 120 µm below the pial surface. (B) Responses of one neuron in (A) to pure tones presented at different frequencies and sound levels. Top panel: single trial (gray) and trial-averaged (black) $\Delta F/F_0$ traces, measured for 1 s from sound onset. Bottom panel: Frequency Response Area (FRA) of the same neuron. Color scale indicates the trial-averaged response of the neuron to tones presented at each frequency/level combination, calculated from the deconvolved fluorescence trace (see Materials and methods). (C) Frequency-response profile (i.e. level-averaged tuning curves) of all neurons in (A). Neurons above the dashed black line were significantly modulated by frequency (two-way ANOVA, p<0.05) and are sorted by their Best Frequency (BF). Neurons below the line were not sensitive to sound frequency and are sorted by the p-value of the frequency predictor in the two-way ANOVA (bottom neurons have the largest p-value). (D) Map of the neurons in (A) color-coded according to their BF (see color scale below). Gray dots represent neurons that were not sensitive to tone frequency. Each gray grid is 100 µm².

The online version of this article includes the following figure supplement(s) for figure 1:

**Figure supplement 1.** GCaMP6 expression in ferret auditory cortex.
**Figure supplement 2.** Imaging in layer 2/3 of ferret A1.

32 imaging fields (e.g. *Figure 1A,B*), while presenting pure tones (0.3–41 kHz; 30–90 dB SPL) to the contralateral ear. To ensure that our recordings were carried out in layers 2/3, we imaged neuronal activity at 176 ± 26.83 µm (median ± SD) below the cortical surface (*Figure 1—figure supplement 2*).

## Frequency response organization at the local scale

Across all fields imaged in ferret A1, 693 of 3604 (19.23%) neurons were frequency-sensitive (two-way ANOVA; p<0.05). Neurons often showed a V-shaped Frequency Response Area (FRA; *Figure 1B*), with a clearly defined Best Frequency (BF). In the example field shown in *Figure 1* (300 × 300 µm), 26.5% of neurons were significantly sensitive to tone frequency or frequency/level combinations, and the average BF across all frequency-sensitive neurons (*Figure 1C*) was 14.7 kHz (±0.75 kHz; mean ± SEM). As previously described for A1 of mice (*Bandyopadhyay et al., 2010*; *Rothschild et al., 2010*; *Panniello et al., 2018*), most neurons within this local region of ferret A1 were tuned to a similar BF, but there were some outliers preferring higher or lower BFs (*Figure 1D*). The BF of individual neurons in this field ranged from 2.2 kHz to 19.9 kHz, with 9% of BFs being more than one octave away from the mean.

The practice of assigning a BF is based on the assumption that neurons are tuned to a single preferred frequency, but some neurons can respond strongly to multiple frequencies (*Bizley et al., 2005*; *Hackett et al., 2011*; *Sutter and Schreiner, 1991*). We observed that the FRAs of frequency-sensitive neurons in our ferret A1 dataset could be classified into three broad types: 1) V-, I- or O-shaped FRAs, which all have a clear single BF ('single-peaked neurons'); 2) FRAs with response peaks at two distinct frequencies ('double-peaked neurons'), where one peak is usually substantially stronger than the other; and 3) more complex FRAs, often containing three response peaks and a poorly defined BF. Little is known about the functions of neurons with these different frequency-response profiles, and their local spatial distribution has not been directly investigated, although neurons with multi-peaked FRAs have been reported in the A1 of marmosets (*Kadia and Wang, 2003*), cats (*Sutter and Schreiner, 1991*), rats (*Turner et al., 2005*) and mice (*Winkowski and*

*Kanold, 2013*). We hypothesized that complexity in the shape of FRAs could help explain some of the reported local variability in cortical tonotopy, as it is less clear where double-peaked and complex FRAs should be located on a tonotopic map.

Each frequency-sensitive neuron was classified using an automated algorithm as having a single-peaked, double-peaked or complex FRA (see Materials and methods; *Figure 2A*). In most imaging fields, we observed all three FRA classes, (as illustrated in *Figure 2B*), and the following analyses were restricted to imaging fields containing at least three neurons of each class. Best frequency was derived in the same manner for neurons across all three classes, as the peak of the level-averaged tuning curve (as shown in Figure 1C; see Methods and materials).

Visual inspection of tonotopic maps (*Figure 2B*; *Figure 2—figure supplement 1*) suggested that the BFs of double-peaked (BF$_d$) and complex neurons (BF$_c$) may be more varied within a local field than those of single-peaked neurons (BF$_s$). To quantitatively compare the local tonotopic organization of neurons with single-peaked, double-peaked and complex FRAs, we computed a BF variance metric. This metric was defined as the octave difference per millimeter between the BF of each neuron and the median BF of all other neurons in the imaging field.

Best frequencies varied less within an imaging field among single-peaked neurons (1.34 ± 0.10 octaves; mean ± SEM) than among double-peaked neurons (2.16 ± 0.18 octaves; t-test: t = −4.15, p=4.1×10$^{-5}$) and complex neurons (3.05 ± 0.24 octaves; t = 7.55, p=3.4×10$^{-13}$) (*Figure 2C*). Within double-peaked neurons, the frequency variability of the second peak (peak 2) and BF$_d$ were comparable (t = 1.91, p=0.057). Therefore, tonotopic organization within a local region of ferret A1 was less reliable among neurons with more complex frequency receptive fields.

We also investigated whether the three classes of neurons differed in their response strengths (*Figure 2D*). We found that the average deconvolved calcium response at the best frequency and level combination was stronger in single-peaked neurons compared to either double-peaked neurons (t-test: t = 2.22, p=0.027) or neurons with complex FRAs (t = 4.02, p=6.9×10$^{-5}$). There was no significant difference in response strength between neurons with double-peaked and complex FRAs (t = 1.58, p=0.11). The Fano Factor calculated at the best frequency and level did not significantly differ between neurons in the three FRA classes (single- and double-peaked: t = 0.60, p=0.55; single-peaked and complex: t = 0.638, p=0.52), indicating that responses at BF were equally reliable for neurons with single- and multi-peaked FRAs (*Figure 2E*). As expected, when data from GCaMP6m and GCaMP6f injections were analyzed separately, we found a higher percentage of frequency-sensitive neurons in the GCaMP6m (32.29%), compared to the GCaMP6f dataset (10.81%). Importantly, both indicators showed similar effects of FRA class on local BF variance, response strength and response reliability (*Figure 2—figure supplement 2*), so our main findings are consistent across the two indicators, and data are pooled across all ferrets for our remaining analyses.

Extracellular recordings are known to be biased towards more active neurons, and *Figure 2D* suggests this may bias them towards single-peaked neurons. In addition, complex neurons may be expected to have more widespread dendritic branches, making them more prone to damage during electrode insertion. If either or both of these effects cause microelectrodes to oversample single-peaked neurons, this could explain the smoother tonotopic maps typically described using this technique. To investigate this possibility, we used high channel count microelectrodes to isolate the tone responses of single neurons in layers 2/3 of ferret A1 (*Figure 2—figure supplement 3*; see Materials and methods), and the recorded FRAs were classified in the same way. The anesthetic regime, surgery, and stimuli were similar to our imaging experiments. Although all three FRA classes were observed, the microelectrode recordings yielded a higher proportion of neurons with single-peaked FRAs (Likelihood Ratio Test: χ$^2$ = 68.45, p=1.1×10$^{-16}$) than the imaging experiments, and fewer with double-peaked (χ$^2$ = 22.16, p=2.5×10$^{-6}$) and complex (χ$^2$ = 30.06, p=4.2×10$^{-8}$) FRAs (*Figure 2F*).

Contrary to our two-photon imaging results, we did not find a significant difference between the spike rates of neurons with single- and double-peaked FRAs (t-test: t = 1.29, p=0.20) or between those with single-peaked and complex FRAs (t = 0.05, p=0.96) in our microelectrode recordings (*Figure 2—figure supplement 4*). This may result from a bias towards the most strongly responsive multi-peaked neurons, given the small number of double-peaked and complex neurons measured using microelectrodes. Together, our results suggest that biases towards sampling single-peaked neurons in microelectrode recording studies could lead to estimates of more ordered tonotopy than two-photon calcium imaging in the same cortical region.

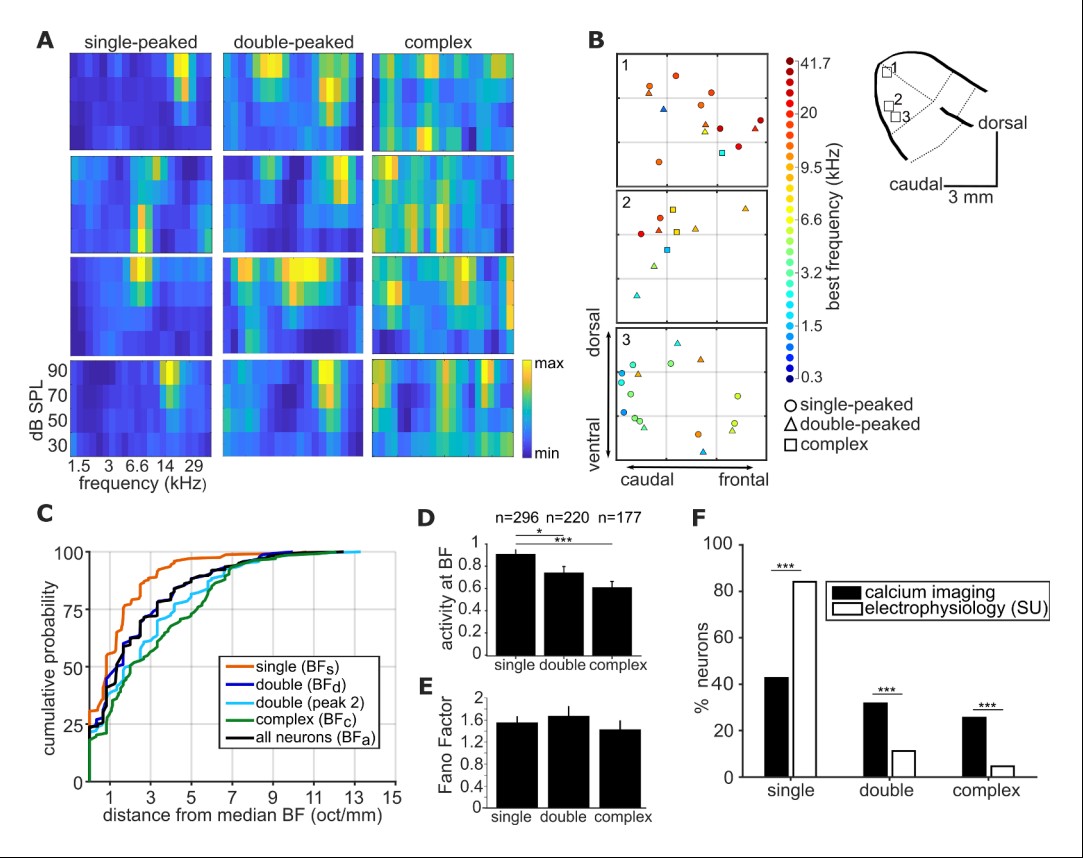

**Figure 2.** Complexity of frequency receptive fields predicts local variability in ferret A1 tonotopy. (**A**) FRA plots of four example neurons from ferret 6 for each of the three FRA classes. (**B**) Three 300 x 300 µm spatial maps of frequency-sensitive neurons along the tonotopic axis of ferret 6. Their anatomical locations within the ferret's ectosylvian gyrus are represented in the schematic to the right. The color of each neuron indicates its BF (see color scale, right), and the shape corresponds to the classification of its FRA. (**C**) Cumulative probability plots of the difference (in octaves) between each neuron's BF and the median BF of all neurons of the same FRA class in the same imaging field. Distributions for different FRA classes are plotted separately: single-peaked $BF_s$; double-peaked $BF_d$; double-peaked peak 2; complex $BF_c$; and the $BF_a$ of all three FRAs together. (**D**) Magnitude of the trial-averaged response at BF, calculated from deconvolved fluorescence traces in two-photon imaging experiments. Responses were averaged across all neurons in each FRA class, pooled across imaging fields and ferrets (mean ± SEM). (**E**) Fano Factor values (mean ± SEM) for single-peaked, double-peaked, and complex neurons. (**F**) Percentage of neurons classified into each FRA class from two-photon calcium imaging (black) and single unit microelectrode recordings (white). The results of t-tests are indicated above the bars in (D) and (E), and Likelihood Ratio Tests in (F) (*p < 0.05 and *** p < 0.001).

The online version of this article includes the following figure supplement(s) for figure 2:

**Figure supplement 1.** Spatial maps of best frequency in A1 of 4 ferrets.

**Figure supplement 2.** BF variance, response strength and response reliability are comparable in GCaMP6m and GCamP6f ferret datasets.

**Figure supplement 3.** Determining cortical layers from an inverse Current Source Density (iCSD) map of electrophysiological responses.

**Figure supplement 4.** Magnitude of the trial-averaged response at the best frequency and level combination, calculated from spike rates measured with microelectrodes.

---

The FRAs of single-peaked neurons can be well approximated by their BF. However, this is not the case for neurons with double-peaked and, in particular, complex FRAs. Consequently, an analysis based on BF alone may be blind to an underlying organization of neurons with complex frequency receptive fields. To take into account a more complete representation of FRAs, we computed pair-wise signal correlations. Correlations were calculated between pairs of frequency-sensitive neurons

imaged simultaneously and belonging to the same FRA category: single-peaked, double-peaked and complex (*Figure 3A*). Signal correlations were found to differ across the three classes (one-way ANOVA: F = 389.70, p=1.6×10$^{-163}$). In *post hoc* pairwise tests (Tukey's HSD), signal correlations within complex neurons (0.048 ± 0.003; mean ± SEM) were significantly lower than those for single-peaked (0.207 ± 0.002; p=9.6×10$^{-10}$) and double-peaked neurons (0.193 ± 0.002; p=9.6×10$^{-10}$). Similarly, signal correlations were significantly higher within single-peaked neurons than double-peaked neurons (p=4.4×10$^{-5}$). Furthermore, signal correlations decreased with cortical distance between neurons for single-peaked (Pearson's correlation: r = 0.21, p=5.3×10$^{-29}$; *Figure 3E*) and double-peaked cells (r = 0.17, p=1.5×10$^{-9}$; *Figure 3F*), but not for complex neurons (r = 0.06; p=0.12; *Figure 3G*). These results further confirm that there is a greater degree of tonotopy for cells with simpler FRAs within a local cortical region.

To investigate whether single- and double-peaked neurons share different local networks from those with complex FRAs, we calculated pairwise noise correlations, which are thought to reflect connectivity and common inputs between neurons (*Hofer et al., 2011*; *Figure 3B*). Indeed, the strength of noise correlations differed across the three FRA classes (one-way ANOVA: F = 109.50, p=9.4×10$^{-48}$). Noise correlations were significantly higher for single-peaked (0.138 ± 0.016; mean ± SEM) and double-peaked (0.136 ± 0.004) neurons than those with complex FRAs (0.059 ± 0.004) (Tukey's HSD tests; single vs complex: p=9.6×10$^{-10}$; double vs complex: p=9.6×10$^{-10}$). Noise correlations did not differ between single-peaked and double-peaked neurons (p=0.86). For all FRA types, noise correlations significantly decreased for pairs of neurons that were located further apart within the imaging field (*Figure 3—figure supplement 1A–C*).

When we examined the correlation structure between neurons with different FRA classifications, we found that single- and double-peaked neurons had higher signal and noise correlations with one

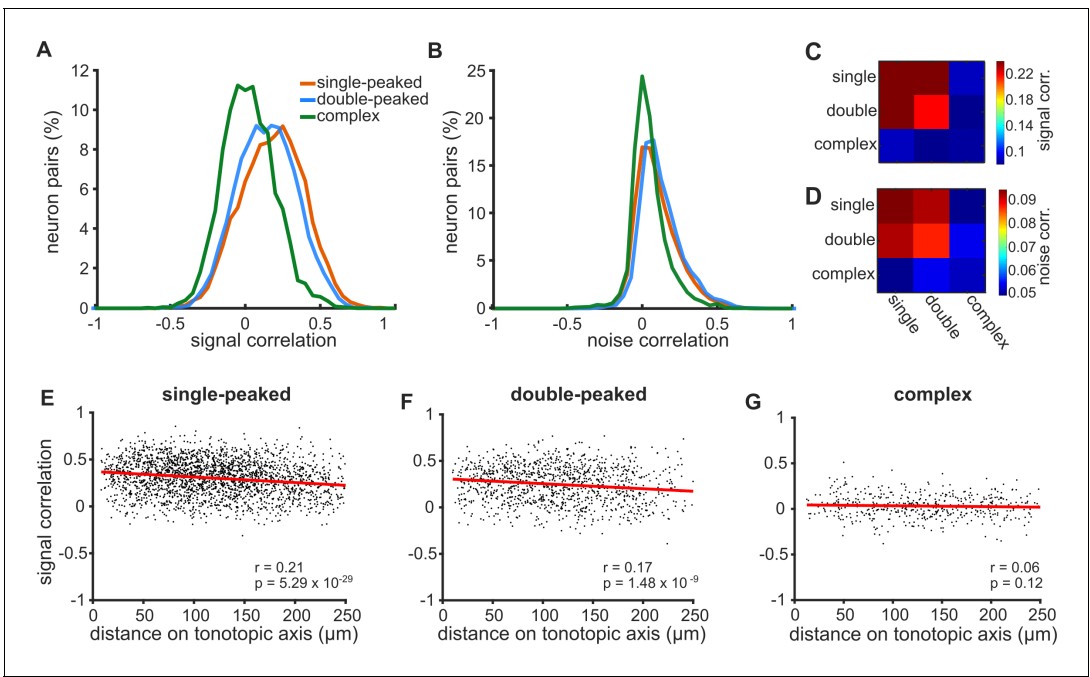

**Figure 3.** Correlations in neural activity are weaker and less spatially ordered for neurons with more complex FRAs. (A) Distributions of signal correlations in the pure tone responses of pairs of simultaneously imaged neurons with single-peaked (orange), double-peaked (blue) or complex (green) FRAs. (B) Distributions of noise correlations for the same neural pairs shown in (A). (C) The color scale (right) shows the average signal correlation for pairs of neurons across all 9 combinations of the three FRA classes. (D) Noise correlations across FRA classes, presented as in (C). (E) Pairwise signal correlations for single-peaked neurons plotted as a function of the distance between the two neurons along the tonotopic axis. The best linear fit to the data is shown (red line), with Pearson's correlation coefficient (r) and p-value (p) in the bottom right of the plot. (F) Same as (E), but for double-peaked neurons. (G) Same as in (E), but for complex neurons.

The online version of this article includes the following figure supplement(s) for figure 3:

**Figure supplement 1.** Noise correlations in neurons with single-peaked, double-peaked and complex FRAs.

another than with complex neurons (*Figure 3C,D*). This suggests that neurons with complex FRAs may differ in their connectivity within the local cortical network from those with simpler frequency tuning.

## Frequency response organization at the global scale

To investigate tonotopic organization along the entire extent of ferret primary auditory cortex, all neurons imaged in our eight ferrets were projected onto a common template of A1. These projections were aligned based on both an anatomical criterion (i.e. the gross anatomy of the ectosylvian gyrus), and a functional one (i.e. A1 boundaries derived from neuronal responses to tones) (*Figure 4—figure supplement 1A,B*; see Materials and methods).

When single-peaked neurons were projected onto this global A1 template, the expected tonotopic gradient was clearly evident (*Figure 4A1*), with high frequencies mapped onto the dorsal tip of the ectosylvian gyrus (*Figure 4B1*) and lower frequencies toward the ventral border with secondary posterior fields (*Kelly et al., 1986a*; *Bizley et al., 2005*). The large-scale spatial organization of frequency preferences along the tonotopic gradient was quantified as a correlation between $BF_s$ and the neuron's position along the tonotopic axis (Pearson's correlation: r = −0.40, p=1.1×10$^{-12}$; *Figure 4C1*).

When examining only neurons with double-peaked FRAs ($BF_d$), global tonotopy was clearly visible (*Figure 4A2, B2*), and this tonotopic gradient was again statistically significant (r = 0.46, p=5.7×10$^{-13}$; *Figure 4C2*). However, the BFs of double-peaked neurons varied more around their fitted tonotopic gradient (red line, *Figure 4C2*) than those of single-peaked neurons (t-test: t = 2.43, p=0.02), suggesting that neurons with double-peaked FRAs have a less smooth tonotopic organization.

The BFs of neurons with complex FRAs were also organized tonotopically on the global A1 template (r = 0.32, p=1.8×10$^{-5}$; *Figure 4A4, B4*), but were more variable around the tonotopic gradient (*Figure 4C4*) than the BFs of either single-peaked (t = 6.83, p=2.7×10$^{-11}$) or double-peaked (t = 4.52, p=8.2×10$^{-6}$) neurons.

In contrast to BF, other aspects of the neurons' frequency responses were not found to be systematically ordered along the tonotopic gradient. The frequency of the second-strongest peak of double-peaked FRAs (peak 2) did not change systematically along this axis (r = −0.096, p=0.15; *Figure 4A3, B3, C3*), nor did the differences between the two peaks of double-peaked neurons (r = 0.13, p=0.046; *Figure 4—figure supplement 2A–C*).

These results indicate that, in ferrets, the BFs of A1 neurons are tonotopically organized at the global scale, but neurons with increasingly complex FRAs show more variance along the global tonotopic axis. In order to validate the method used for deconvolving calcium traces, we repeated our analyses on non-deconvolved $\Delta F/F_0$ traces (*Figure 4—figure supplement 3*). The local BF variability was similar when computed from non-deconvolved (*Figure 4—figure supplement 3A*) and deconvolved traces (*Figure 2C*). In keeping with the data from the deconvolved signals, we found that the average amplitude of the calcium transient at BF was significantly higher in single-peaked neurons compared to double-peaked (t-test: t = 3.59, p=3.5×10$^{-4}$) and complex neurons (t = 5.10, p=5.0×10$^{-7}$) (*Figure 4—figure supplement 3B*; compare to *Figure 2D*). In addition, the Fano Factor calculated at BF from the non-deconvolved traces did not significantly differ between the three FRA classes (single- and double-peaked: t = 0.25, p=0.80; single-peaked and complex: t = 1.030, p=0.30; *Figure 4—figure supplement 3C*; compare to *Figure 2E*). At the global scale, the tonotopic organization was evident when BF was calculated from non-deconvolved traces (*Figure 4—figure supplement 3D*; compare to *Figure 4B*), and BF variability along the tonotopic axis was similar in the deconvolved and non-deconvolved datasets (two-way ANOVA: F = 1.24, p=0.27; *Figure 4—figure supplement 3E*).

Our analysis of both the local and global organization of frequency responses was carried out after removing the neuropil signal from each individual neuronal body. This procedure was necessary, as neuropil contamination can change the BF of individual neurons, particularly for those with complex frequency responses (see Materials and methods; *Figure 4—figure supplement 4A,B*), as also highlighted in a recent study of mouse tonotopy (*Romero et al., 2020*). For example, we found that neuropil signal introduced the appearance of a tonotopic order to an otherwise spatially disorganized signal – namely, the second peaks of double-peaked neurons (*Figure 4—figure supplement 4D3*).

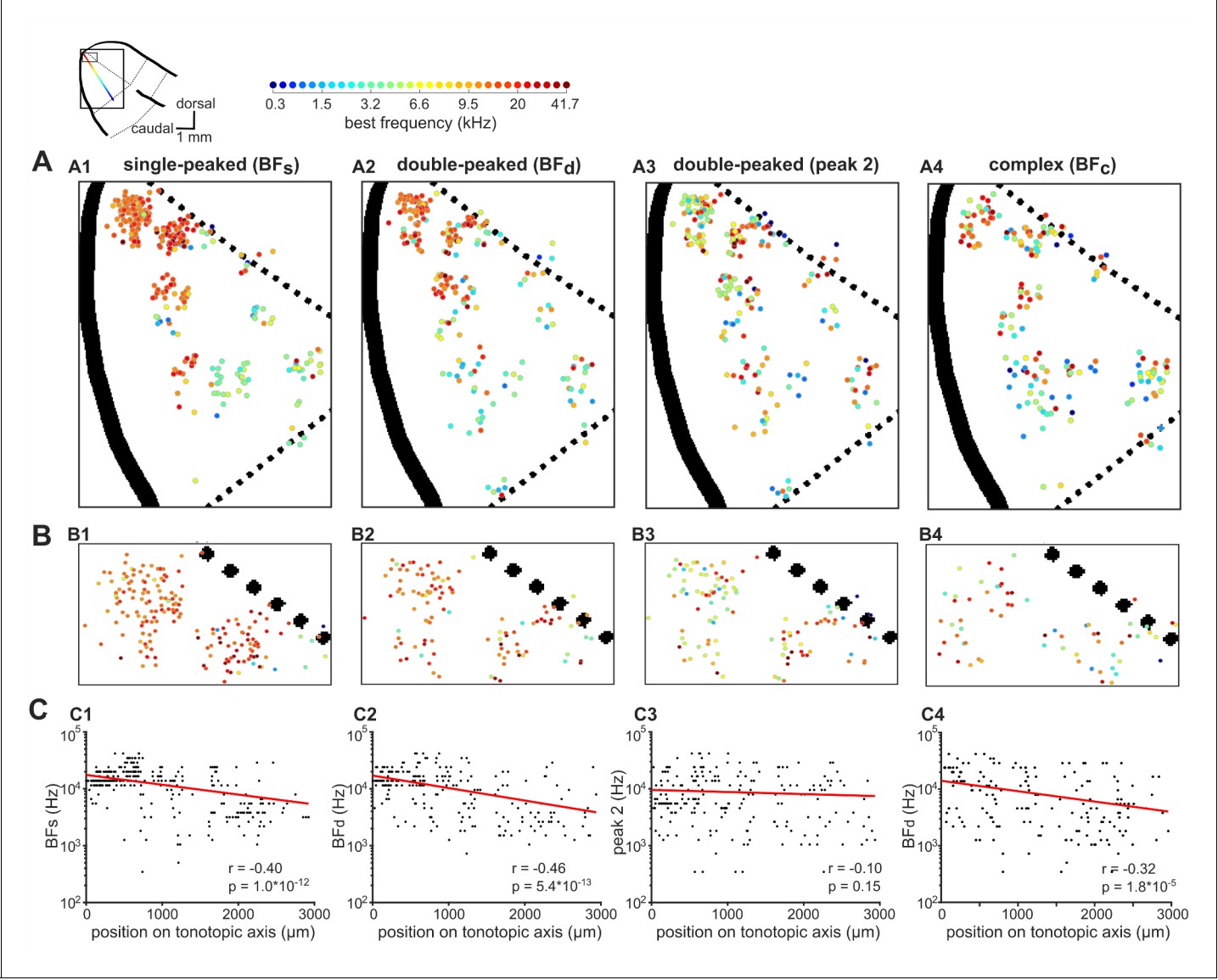

**Figure 4.** Global tonotopic organization of frequency preferences in ferret A1. The anatomical locations of neurons imaged across eight ferrets were projected onto a template map of auditory cortex, shown in the top-left corner. Thick black solid lines indicate sulci, and black dotted lines indicate approximate borders between known cortical fields. The two boxes represent the location of A1 (large box; **A**) and the most dorsal region of A1 (small box; **B**). The colored line illustrates the tonotopic gradient. (**A**) The BFs of individual neurons are color-coded (legend above) and mapped onto A1. The spatial distributions of $BF_s$ (A1), $BF_d$ (A2), peak 2 of double-peaked neurons (A3), and $BF_c$ (A4) are plotted separately. (**B**) Frequency preferences are mapped as in (**A**) for the dorsal tip of A1, where many neurons are occluded in (**A**). (**C**) $BF_s$ (C1), $BF_d$ (C2), peak 2 (C3), and $BF_c$ (C4) of each neuron are plotted against the neuron's position along the tonotopic axis on the template A1. Red lines show the best single-term exponential fits to the data, and Pearson's correlations (r) with their p-values (p) are also shown.

The online version of this article includes the following figure supplement(s) for figure 4:

**Figure supplement 1.** Mapping data from individual ferrets onto a common template of auditory cortex.

**Figure supplement 2.** Mapping the peak-to-peak distances in double-peaked neurons.

**Figure supplement 3.** Comparison between deconvolved and non-deconvolved traces.

**Figure supplement 4.** Effects of neuropil contamination on local and global tonotopic organization.

## Comparison of tonotopic organization in ferrets and mice

The contrast between local heterogeneity and global tonotopic organization of BFs, sometimes referred to as 'fractured' or 'salt and pepper' organization, has been previously reported in the primary auditory cortex of the mouse (*Bandyopadhyay et al., 2010*; *Rothschild et al., 2010*;

*Panniello et al., 2018*). The results above show that a qualitatively similar organization exists in ferret A1, but that much of the heterogeneity at both the local and global scales arises from neurons with complex frequency receptive fields. To directly compare the local organization of frequency preferences between mice and ferrets, we imaged neuronal responses in the primary auditory cortex of 11 mice under the same experimental conditions.

The BFs of neurons and their spatial organization within an example imaging field in the mouse are shown in *Figure 5A and B*, respectively. In keeping with previous studies (*Bandyopadhyay et al., 2010*; *Rothschild et al., 2010*), these plots suggest that the spatial organization of BFs in the mouse primary auditory cortex is also locally heterogeneous. Across all imaging fields, 854 of 1964 (43.75%) neurons were frequency-sensitive in mice (two-way ANOVA; $p < 0.05$), while only 693 of 3604 (19.23%) neurons were frequency-sensitive using the same statistical criterion in ferrets.

Classification of neurons based on their FRAs confirmed that single-peaked, double-peaked, and complex neurons also exist in the mouse (*Figure 5B*; *Figure 5—figure supplement 1A–B*), as they do in the ferret (*Figure 2B*). However, the relative proportions of these three FRA classes differ between the two species (*Figure 5C*). Specifically, a higher percentage of frequency-sensitive neurons in the mouse (60.54%) showed 'simpler' frequency receptive fields with a single peak at BF than

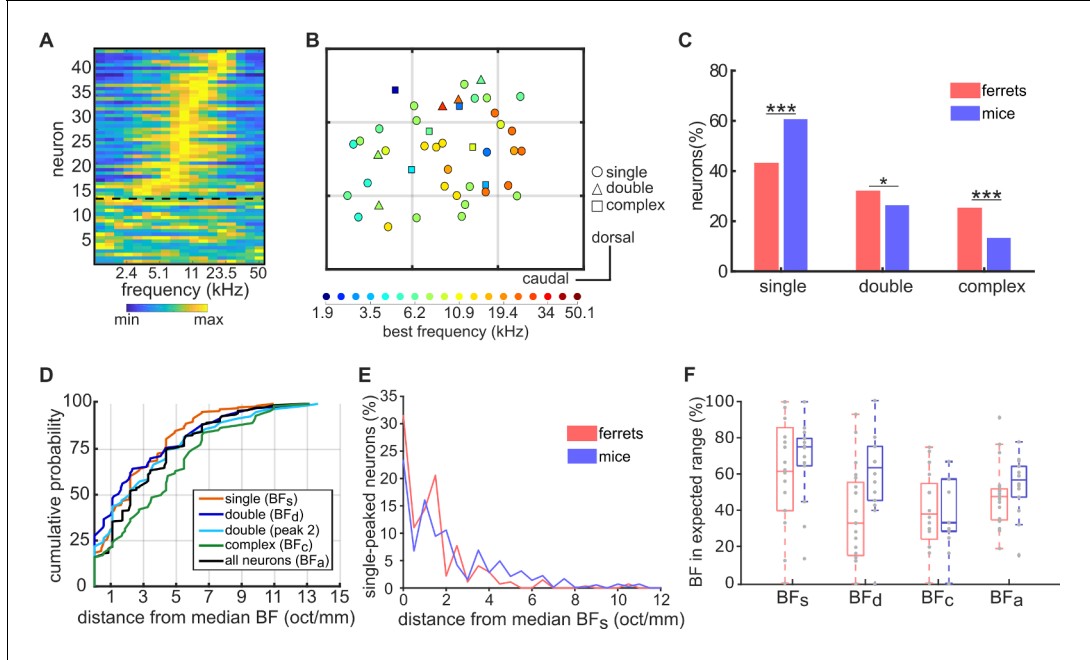

**Figure 5.** Similar tonotopic organization in the primary auditory cortex of ferrets and mice. (A) Frequency-response profile (i.e. level-averaged tuning curves) of all neurons imaged in one imaging field in one mouse. Data are presented as in *Figure 1C*. (B) Map of the anatomical locations of the neurons in (A), plotted as in *Figure 2B*. Neurons are color-coded according to their BF, and their shape corresponds to their FRA class. Each gray grid corresponds to a cortical area of 100 x 100 μm. (C) Proportion of frequency-sensitive neurons from each FRA class for ferrets (red) and mice (blue). The results of Likelihood Ratio tests are indicated above (*p < 0.05 and *** p < 0.001). (D) Cumulative probability plots of the difference (octaves/mm) between the BF of each neuron and the median BF of all neurons in the same imaging field. As in *Figure 2C*, distributions were calculated separately for BF$_s$ (orange), BF$_d$ (dark blue), peak 2 of double-peaked neurons (light blue), BF$_c$ (green), and BF$_a$ (black). (E), Distributions of the distance in octaves per mm between the BFs of each single-peaked neuron in an imaging field and the median BF of these neurons. Data are shown for ferrets (red) and mice (blue). (F) Box plots showing the percentage of neurons having a BF within the expected frequency range, given the species, size and average BF of each imaging field. Percentages were calculated separately for BF$_s$, BF$_d$, BF$_c$ and BF$_a$, and percentages for individual imaging fields are shown as gray dots. Boxplots for ferrets (red) and mice (blue) show the median, upper quartile, and lower quartile of each dataset.

The online version of this article includes the following figure supplement(s) for figure 5:

**Figure supplement 1.** All 3 FRA types are present in the mouse primary auditory cortex.

**Figure supplement 2.** FRA bandwidth does not predict local tuning heterogeneity.

**Figure supplement 3.** The BF variance within an imaging field does not depend on the number of neurons in the field.

**Figure supplement 4.** Effects of FRA complexity on BF organization hold across all imaged neurons.

in the ferret (42.71%; Likelihood Ratio Test: $\chi^2$ = 48.96, p=2.6×10$^{-12}$). Conversely, ferrets had more double-peaked (31.75% vs 26.23%; $\chi^2$ = 5.67, p=0.017) and complex FRAs (24.96% vs. 13.23%; $\chi^2$ = 34.83, p=3.6×10$^{-9}$) than mice. Single-peaked and double-peaked neurons in mice responded more strongly at BF than complex neurons (single- and double-peaked: t = −0.73, p=0.46; single-peaked and complex: t = 3.20, p=0.0010; double-peaked and complex: t = 3.43, p=1.9×10$^{-4}$; *Figure 5—figure supplement 1C*), and trial-to-trial reliability in complex neurons was better than that of single-peaked neurons (single- and double-peaked: t = −0.99, p=0.32; single-peaked and complex: t = 2.37, p=0.018; *Figure 5—figure supplement 1D*).

Neurons with single-peaked FRAs showed broader frequency tuning in mice (0.91 ± 0.03 octaves) compared to ferrets (0.74 ± 0.03 octaves; t-test: t = 3.66, p=2.7×10$^{-4}$). This could partially account for the larger proportion of single-peaked neurons in mice, if the dominant peak occupies a greater proportion of the maximum bandwidth. However, there were no significant species differences in tuning bandwidth for double-peaked (t = 0.76, p=0.44) or complex neurons (t = 0.72, p=0.47; *Figure 5—figure supplement 2A*). Furthermore, larger bandwidths of single-peaked neurons were not associated with larger deviations in local BF in ferrets (Pearson's correlation: r = 0.020, p=0.73; *Figure 5—figure supplement 2B*), and were weakly correlated with smaller BF deviations in mice (r = −0.11, p=0.030; *Figure 5—figure supplement 2C*). In both species, bandwidth and BF variability were not significantly correlated in either double-peaked or complex neurons (p>0.05). Therefore, the sharpness of frequency tuning does not explain local BF variability.

Could the presence of double-peaked and complex FRAs help explain the local variance in cortical tonotopy in mice, as we observed in ferrets? To answer this question, the BF variance within an imaging field was calculated separately for single-peaked, double-peaked and complex neurons (*Figure 5D*), in every imaging field containing at least three neurons of each FRA class. Unlike in ferrets, the BF variability of double-peaked neurons (2.73 ± 0.33 oct/mm; mean ± SEM) was not significantly higher than that of single-peaked neurons (2.57 ± 0.16 oct/mm; t-test: t = 0.46, p=0.64) in mice, and the BF variability of the two peaks of double-peaked neurons did not differ significantly (t = 1.15, p=0.25). In line with our ferret data, however, the BFs of neurons with complex FRAs (4.14 ± 0.38 oct/mm) were more variable than those of single-peaked (t = 4.33, p=2.0×10$^{-5}$) and double-peaked (t = 2.78, p=0.0060) neurons, suggesting that the frequency preferences of neurons with more complex FRAs are less tonotopically organized within the local microcircuit. To ensure that the smaller number of multi-peaked neurons in mice did not bias our estimates of BF variance, we repeated this analysis using the same number of neurons in each imaging field. Furthermore, we bootstrapped our results across different subgroups of neurons in each field. The same patterns of BF variance were found across single-peaked, double-peaked and complex cells in these control analyses (*Figure 5—figure supplement 3B,C,E and F*).

As many neurons in sensory cortex can be unresponsive or poorly tuned to pure tone frequency, it is common practice to map only neurons that are significantly modulated by sound frequency (*Rothschild et al., 2010*; *Tischbirek et al., 2019*). However, this inclusion criterion may favor the emergence of smoother BF distributions in mapping studies. Therefore, we repeated our BF analyses on all imaged neurons. The inclusion of more noisy neurons in this analysis predictably resulted in a larger proportion of complex FRAs overall. Importantly, mice again showed substantially more single-peaked neurons than ferrets (*Figure 5—figure supplement 4A*), as in our analysis of frequency-sensitive neurons (*Figure 5C*). Furthermore, more scatter in the spatial distribution of BF was observed with increased FRA complexity at both local (*Figure 5—figure supplement 4B and C*) and global (*Figure 5—figure supplement 4D–G*) levels of tonotopic organization, as we found for frequency-sensitive neurons (*Figure 2C*; *Figure 4C*; *Figure 5D*).

A comparison of the tonotopic organization between mice and ferrets is complicated by the anatomical and hearing range differences between the two species. The A1 tonotopic gradient is shorter in the mouse (~1 mm) (*Guo et al., 2012*; *Stiebler et al., 1997*) than the ferret (~3.5 mm) (*Kelly et al., 1986a*; *Bizley et al., 2005*), and the audible hearing range is typically 2–60 kHz in C57BL/6 mice (*Heffner and Heffner, 2007*; *Ison et al., 2007*) and 0.3–44 kHz in ferrets (*Kelly et al., 1986b*). To illustrate the importance of these species differences, *Figure 5E* compares histograms of the BF$_s$ variance in ferrets and mice, expressed in units of octaves per mm. The BFs of single-peaked neurons were more variable within a millimetre of A1 in mice (2.18 ± 0.08 octave/mm) than in ferrets (1.33 ± 0.09; t-test: t = 5.62, p=2.5×10$^{-8}$). This may be taken to suggest that BF$_s$ is more locally

variable in mouse primary auditory cortex than in ferrets, but this metric does not account for the species differences in hearing range and tonotopic axis length.

To account for this difference, we computed the percentage of neurons that had a BF within the expected frequency range in each species, for a given imaging field. The expected BF range was calculated based on the diameter of the imaging field, the median BF across all neurons in the field, and an expected tonotopic gradient of 4.91 oct/mm in mice and 2.93 oct/mm in ferrets. To simplify the calculation, this measure assumes a constant log frequency gradient across the entire tonotopic axis. The percentage of neurons with a BF within the expected range provided a measure of the variance in tonotopic organization that could be directly compared between species, and was computed separately for neurons with: 1) single-peaked FRAs; 2) double-peaked FRAs; 3) complex FRAs; and 4) all three FRA types combined (*Figure 5F*). The resulting values varied across the three different FRA classes (two-way ANOVA: $F = 8.41$, $p=4.3\times10^{-4}$), but not between species ($F = 3.72$, $p=0.057$), and there was no significant interaction between FRA class and species ($F = 1.66$, $p=0.19$). Furthermore, *post hoc* pairwise Tukey's HSD tests found no significant differences between BF variability in mice and ferrets for single-peaked ($p=0.97$), double-peaked ($p=0.12$), or complex neurons ($p=1.0$).

Taken together, these results suggest that mice and ferrets have equivalent local heterogeneity in frequency preferences within each FRA class of neurons in layers 2/3 of primary auditory cortex. More complex frequency tuning is associated with more variance in the frequency preferences of neighboring neurons in both species.

## Discussion

We used two-photon calcium imaging to study the representation of sound frequency in populations of neighboring neurons in mouse and ferret primary auditory cortex. Previous studies in mice have described global A1 tonotopy with local variability of frequency preferences (*Bandyopadhyay et al., 2010*; *Rothschild et al., 2010*; *Panniello et al., 2018*; *Romero et al., 2020*; *Winkowski and Kanold, 2013*; *Tischbirek et al., 2019*), and here we show that ferret A1 shares the same spatial organization. Furthermore, we demonstrate that the complexity of frequency receptive fields can account for much of the local heterogeneity in both mice and ferrets. Thus, the locally heterogeneous sensory maps described in primary cortical areas may be a consequence of complex stimulus feature extraction.

### Primary auditory cortex is tonotopically organized at the global scale

Electrophysiological and widefield imaging studies have described a tonotopic gradient in the primary auditory cortex of all mammalian species studied, including humans (*Romani et al., 1982*), monkeys (*Merzenich and Brugge, 1973*), cats (*Hind, 1953*), ferrets (*Kelly et al., 1986a*), gerbils (*Steffen et al., 1988*), rats (*Sally and Kelly, 1988*) and mice (*Stiebler et al., 1997*). This global tonotopic organization has been confirmed in two-photon calcium imaging studies of primary auditory cortex in mice (*Bandyopadhyay et al., 2010*; *Rothschild et al., 2010*; *Romero et al., 2020*; *Winkowski and Kanold, 2013*; *Tischbirek et al., 2019*; *Issa et al., 2014*) and marmosets (*Zeng et al., 2019*). Here, we performed the first two-photon calcium imaging of auditory cortex in carnivores, and show that this result generalizes to ferrets. The stereotyped position of ferret A1 within reliable anatomical landmarks allowed us to combine data from multiple animals onto a template of A1, as in previous electrophysiological studies (*Bizley et al., 2009*). This map showed a clear dorsal-to-ventral tonotopic gradient for neurons with single-peaked, double-peaked and, to a lesser extent, complex frequency receptive fields. In contrast, we observed no systematic spatial arrangement of the second frequency peak, or in the difference between frequency peaks in double-peaked neurons. Together, these findings support the well-established idea that global tonotopy is the main spatial organizational principle in primary auditory cortex, and is observed within all neurons that are frequency sensitive.

### Tonotopy is well preserved locally for neurons with simple, but not complex, frequency receptive fields

An advantage of two-photon calcium imaging over other techniques is its ability to sample the activity of large numbers of individual neurons within a local cortical region, while knowing their relative spatial locations. In both ferrets and mice, neurons with single-peaked FRAs were organized

homogeneously, with almost 90% of them in a given imaging field presenting BFs within the expected frequency range, assuming a tonotopic gradient of < 4.9 oct/mm in mice and 2.9 oct/mm in ferrets. Signal correlation analysis further confirmed local tonotopic organization, as proximate neurons were more similarly tuned than distant ones within an imaging field (~250 μm²). Double-peaked neurons were less ordered, with ~ 60% of their BFs within the expected range and greater local variability around the global tonotopic gradient, compared to single-peaked neurons. The BFs of neurons with complex frequency responses were even more spatially varied than double-peaked neurons. In contrast to single- and double-peaked neurons, only 30% of complex neurons in a given imaging field were within the expected BF range, and their signal correlations were lower and less spatially dependent. Previous studies have identified multi-peaked neurons in the A1 of mice (*Winkowski and Kanold, 2013*), rats (*Turner et al., 2005*), cats (*Sutter and Schreiner, 1991*) and marmosets (*Kadia and Wang, 2003*), and Romero et al. have recently shown that BFs are more locally variable among neurons with poorly-defined frequency tuning (*Romero et al., 2020*). Here, we further show that BFs are also more variable among nearby neurons with complex frequency receptive fields, even when their trial-to-trial responses at BF are equally reliable (as shown by our Fano Factor analysis). The association between frequency receptive field complexity and tonotopy was remarkably similar for mice and ferrets, suggesting that this may be a common feature of the mammalian cortex. The complexity of receptive fields may also explain the observed local variability of orientation tuning in V1 (*Bonin et al., 2011*; *Andermann et al., 2013*) and whisker preferences in barrel cortex (*Kerr et al., 2007*; *Sato et al., 2007*).

The departure from strict tonotopy in A1, compared to earlier processing stations, such as the central nucleus of the inferior colliculus (*Barnstedt et al., 2015*), may in fact be an unavoidable result of complex spectrotemporal processing. We found that noise correlations were lower within local populations of complex neurons than within single- and double-peaked neurons. This may be due to neurons with complex frequency responses making more horizontal connections with distant cortical columns, or receiving inputs from layer 4 neurons in different areas of the tonotopic map. In V1, synaptic connections are more probable among neurons with similar orientation tuning (*Cossell et al., 2015*). From our experiments, we might expect A1 neurons with simple frequency tuning to have stronger local connections to similarly tuned neurons, supporting well-defined tonotopy. Those with complex frequency responses, on the other hand, may receive inputs from neurons in more distant regions of the tonotopic gradient.

## Similarities and differences between mice and ferrets

Our results show that an equivalent tonotopic organization of well-tuned neurons exists in ferrets and mice, despite the fact that the A1 of ferrets represents ~ 3 oct/mm, while that of mice represents ~ 5 oct/mm. Thus, local heterogeneity in the tonotopic map is unlikely to be due to limitations in brain size, or other potential factors that are inherent to rodents.

In contrast to this view, *Zeng et al., 2019* recently reported that BFs of A1 neurons are more locally variable in rats than in marmosets. It is possible that the marmoset cortex contains a smoother tonotopic arrangement than other species studied, and this would be consistent with the high number of single-peaked neurons presented in their study (*Zeng et al., 2019*). Alternatively, the reported difference between marmoset and rat A1 tonotopy might be accounted for if differences in the size of A1, complexity of frequency tuning, and hearing ranges of these two species are taken into consideration.

*Issa et al., 2014* reported more homogenous tonotopy in mouse A1 than other two-photon imaging studies (*Bandyopadhyay et al., 2010*; *Rothschild et al., 2010*; *Panniello et al., 2018*; *Romero et al., 2020*; *Tischbirek et al., 2019*), although the local variability in BF was still higher than that of marmosets (*Zeng et al., 2019*). Two distinguishing methodological features of *Zeng et al., 2019* and *Issa et al., 2014* are: (1) Issa et al. imaged tone responses in awake animals; and (2) neither study applied neuropil corrections. Both two-photon calcium imaging (*Romero et al., 2020*; *Tischbirek et al., 2019*) and single unit electrophysiological studies (*Guo et al., 2012*) have demonstrated that there is little change in a neuron's BF between passively listening and anesthetized states, so this alone is unlikely to account for the smooth tonotopy observed by *Issa et al., 2014*. Neuropil signals contaminating neural responses can lead to smoother tonotopic maps (*Romero et al., 2020*), on the other hand, as the signals in axons and dendrites are averaged across the local population. In fact, our comparison of data with and without neuropil correction (*Figure 4—*

*figure supplement 3*) demonstrated that neuropil contamination can result in the second, weaker peak of double-peaked neurons appearing to be tonotopically organized, while this is not spatially ordered in the neuropil-corrected signal. Finally, Issa et al. used the genetically encoded calcium indicator GCaMP3, which has a lower calcium sensitivity than GCaMP6. This may have led to the detection of only the strongest variations in calcium influx, typically seen here in single-peaked neurons.

Our study highlighted two main differences between mice and ferrets. First, in both species, the majority of neurons were not sensitive to changes in tone frequency, but fewer A1 neurons exhibited frequency sensitivity in ferrets (19%) than in mice (44%). While these numbers may seem low, most superficial A1 neurons do not even respond to pure tones, especially in anesthetized animals (*Panniello et al., 2018*; *Tischbirek et al., 2019*). Therefore, it is important to keep in mind that tonotopic maps do not reflect the spatial organization of response properties of most auditory cortical neurons. Second, a greater proportion of frequency-sensitive neurons showed double-peaked or complex FRAs in ferrets than in mice. In contrast, the proportions of multi-peaked neurons in our microelectrode recordings in ferret A1 are equivalent to those reported in marmoset A1 recordings (*Kadia and Wang, 2003*). These results may indicate that neurons in ferret and marmoset A1 are better suited to integrate information across frequency bands or encode more complex spectrotemporal features. For example, neurons with multiple frequency peaks would be useful for identifying sound sources based on their spectral timbre (*Bizley et al., 2009*), or may provide harmonic templates for pitch perception (*Feng and Wang, 2017*). Further physiological and behavioural studies are required to test these potential species differences in auditory cortical function.

While layers 2 and 3 are often regarded as a single processing unit, neurons in these two laminae of A1 differ in their morphology, connectivity and function (*Atencio and Schreiner, 2010*; *Oviedo et al., 2010*; *Meng et al., 2017*). In our study, most ferret imaging areas were likely located in layer 2 (based on their depth), while calcium transients in mice were imaged in both layers 2 and 3. This may contribute to the species differences in our results. For example, Meng et al reported that neurons in layer 2 have broader frequency tuning than those in layer 3 (*Meng et al., 2017*). This potential laminar difference in imaging location could partially explain why neurons imaged in our mouse dataset had broader frequency tuning than our ferret data.

## Comparison to previous electrophysiological studies

With some exceptions (*Bizley et al., 2005*), electrophysiological studies have typically reported smoother tonotopic gradients than two-photon imaging studies (*Hackett et al., 2011*; *Stiebler et al., 1997*). Our results offer an explanation for this inconsistency across methodologies. Compared to neurons with more complex FRAs, the BFs of single-peaked neurons are more precisely mapped at both the local and global scale, and they also have stronger responses, as measured through their calcium dynamics (*Figure 2D*). Because extracellular microelectrode recordings are more likely to detect and cluster neurons with higher firing rates (*Shoham et al., 2006*), such studies may oversample neurons with single-peaked FRAs and therefore with more precise tonotopy. In support of this hypothesis, we found that > 80% of frequency-sensitive neurons that we recorded from superficial A1 using microelectrodes had single-peaked FRAs: approximately double the proportion of single-peaked FRAs identified using two-photon calcium imaging in the same species. An electrophysiological study of marmoset auditory cortex also found that 20% of neurons showed multi-peaked frequency tuning (*Kadia and Wang, 2003*).

Electrophysiological studies in mice have reported that the thalamorecipient layers of A1 are more tonotopically ordered than more superficial layers (*Hackett et al., 2011*; *Guo et al., 2012*; *Stiebler et al., 1997*). However, two-photon imaging studies have provided conflicting evidence about this (*Winkowski and Kanold, 2013*; *Tischbirek et al., 2019*). We suggest that the more ordered tonotopy in layers 4/5 may arise from a larger proportion of single-peaked neurons in this part of the cortex. This is consistent with the electrophysiological data of *Guo et al., 2012*, who reported that more neurons with irregular frequency receptive fields exist in the superficial layers of mouse A1. Here, we show that tonotopic variability in layers 2/3 is not simply due to neurons having less reliable responses, but is also associated with neurons that respond to a combination of distinct frequency bands.

## Conclusions

Applying two-photon imaging to the study of neuronal activity in the ferret auditory cortex for the first time, we found that A1 neurons tuned to a single best frequency are tonotopically organized at both the local and the global scales. The presence of neurons with more complex FRAs disrupts this strict tonotopic order, increasing the local heterogeneity of neuronal frequency preferences, just as it does in mice. Cells with complex sensory receptive fields are likely to be important for extracting information from natural environments, and are more common in ferrets than mice. Future imaging studies are required to better understand the functional properties and connectivity of these subpopulations of neurons.

# Materials and methods

## Animals

All animal procedures were approved by the local ethical review committee of the University of Oxford and performed under license from the UK Home Office. Eight female ferrets (*Mustela putorius furo*; Marshall BioResources, UK) and 11 female mice (C57BL/6J; Harlan Laboratories, UK) were used in the two-photon imaging experiments, and six ferrets (two male) were used in the electrophysiology experiments.

## Viral vector injections

### Ferret surgery

At age 8–12 weeks, ferrets were put under general anesthesia with an intramuscular injection of ketamine (Vetalar; 5 mg/kg) and medetomidine (Domitor; 0.02 mg/kg), and medicated with buprenorphine (Vetergesic; 0.02 mg/kg i.m.), atropine (Atrocare; 0.06 mg/kg i.m.) and meloxicam (Metacam; 0.2 mg/kg i.m.). Ferrets were then intubated and artificially ventilated. A mixture of oxygen and isoflurane (IsoFlo; 0.5–2%) was continuously delivered throughout the surgery to maintain anesthesia. Respiratory rate, end-tidal $CO_2$, electrocardiogram, blood pressure, and blood oxygenation were continuously monitored, and body temperature was maintained at 36–38°C (3M Bair Hugger). The eyes were lubricated (Maxitrol, Alcon) to prevent corneal desiccation. An intravenous cannula was inserted to deliver Hartmann's solution (54 ml/kg/hr) continuously, a single dose of co-amoxiclav (Augmentin, 20 mg/kg), and medetomidine (Domitor; 0.027 mg/kg i.m.) as required, throughout the surgery. After placing the ferrets in a stereotaxic frame (Model 900LS, David Kopf Instruments), the scalp was cleaned (ChloraPrep; 2% chlorhexidine gluconate), bupivacaine (Marcain; 2 mg/kg, s.c.) was injected into the scalp, the scalp was incised, and the right temporal muscle was retracted. A craniotomy and durotomy were carried out to expose the primary and secondary auditory cortex, based on stereotaxic coordinates (11 mm ventral to the midline and 8 mm frontal to Lambda) and visual confirmation of the ectosylvian gyrus.

Injections were carried out using a glass pipette and a custom-made pressure injection system. The viral vector consisted of a 1:1 solution of AAV1.Syn.GCaMP6m.WPRE.SV40 (Penn Vector Core) and phosphate-buffered saline (PBS; Sigma Aldrich) for ferrets 1–4, and a 1:1 solution of AAV1.Syn.GCaMP6f.WPRE.SV40 (Addgene) and AAV1.mDlx.GFP-GCaMP6f-Fishell-2.WPRE.SV40 (Penn Vector Core) for ferrets 5–8. Approximately 260 nl of the viral vector was slowly injected at depths of ∼ 800, 600, 400 and 200 μm below the pial surface in ferrets 1–4 and at depths of 200 μm and 400 μm in ferrets 5–8. For each ferret, the viral vector solution was injected into 4–9 sites across primary and secondary auditory cortex.

The bone was then replaced over the craniotomy, the muscle, fascia and skin were sutured closed, and the animal was recovered. Dexamethasone (0.5 mg/kg i.m.) and buprenorphine (0.01 mg/kg i.m.) were administered immediately after surgery. Meloxicam (0.2 mg/kg oral) and co-amoxiclav (20 mg/kg s.c.) were administered daily for 5 days, and buprenorphine (0.01 mg/kg i.m.) for two days.

### Mouse surgery

Mouse experiments were previously reported in *Panniello et al., 2018*. At age 5–6 weeks, mice were premedicated with dexamethasone (Dexadreson; 4 μg s.c.), atropine (Atrocare; 1 μg s.c.) and

carprofen (Rimadyl; 0.15 µg s.c.), and put under general anesthesia with fentanyl (Sublimaze; 0.05 mg/kg i.p.), midazolam (Hypnovel; 5 mg/kg i.p.), and medetomidine hydrochloride (Domitor; 0.5 mg/kg i.p.). The mouse was then placed in a stereotaxic frame (Model 900LS, David Kopf Instruments) and maintained at 36–37°C body temperature throughout surgery (DC Temperature Controller, FHC). The scalp was incised, the temporal muscle retracted, and A1 was located using stereotaxic coordinates (70% of the distance from Bregma to Lambda, and ~ 4.5 mm lateral from the midline). Two small holes (~0.4 mm diameter), separated rostrocaudally by ~0.5 mm, were drilled over the right A1. A total of ~ 200 nl of the viral construct AAV1.Syn.GCaMP6m.WPRE.SV40 (Penn Vector Core), diluted (1:2) in PBS, was injected at each site, spread equally across four depths spanning 50–400 µm below the pial surface. After injection, the skin was sutured and general anesthesia was reversed with flumazenil (Anexate; 0.5 mg/kg s.c.) and atipamezol (Antisedan; 2.5 mg/kg s.c.). Postoperative buprenorphine (Vetergesic; 1 ml/kg s.c.) and enrofloxacine (Baytril; 2 ml/kg s.c.) were administered immediately after surgery and again 24 hr later. To confirm that we were targeting mouse A1 with our viral vector injections, the coordinates used matched those of fluorescent retrobead injections that retrogradely labelled neurons in the ventral division of the medial geniculate body (*Panniello et al., 2018*). These injection sites also correspond to the location of A1 determined by imaging the full tonotopic gradient over larger cortical areas of transgenic mice expressing GCaMP6f in cortical neurons under the CaMKII promoter (*Panniello et al., 2018*). Nevertheless, it is not possible to say with the same degree of confidence as for the ferret data that all mouse imaging reported in the present study was performed in A1.

## In vivo two-photon calcium imaging
### Ferret surgery
In vivo two-photon imaging was performed 3–6 weeks after viral injection. General anesthesia was induced with an intramuscular injection of ketamine (Vetalar; 5 mg/kg) and medetomidine (Domitor; 0.02 mg/kg), and was maintained with a continuous intravenous infusion of these two drugs in Hartmann's solution with 3.5% glucose and dexamethasone (0.5 mg/ml/hr). The animal was intubated and artificially ventilated. Respiratory rate, end-tidal $CO_2$, electrocardiogram and blood oxygenation were continuously monitored throughout the imaging session. Eye ointment (Maxitrol; Alcon, UK) was applied throughout and body temperature was maintained at 36–38°C. Atropine (Atrocare; 0.06 mg/kg i.m.) was administered every 6 hr, or when bradycardia or arrhythmia was observed.

Ferrets were placed in a custom-built stereotaxic frame and head stability was achieved using ear bars and a mouthpiece. After shaving the scalp and injecting bupivacaine (Marcain, 2 mg/kg s.c.), the skin was incised and the temporal muscle removed. A steel holding bar was secured to the skull using dental cement (SuperBond; C and B, UK) and a stainless steel bone screw (Veterinary Instrumentation, UK). A circular craniotomy 10 mm in diameter was drilled over the injection site, and any dura that had regrown over the auditory cortex was removed. A custom made titanium ring (10 mm diameter) containing an 8 mm glass coverslip (Thermo Fisher Scientific, UK) was inserted in the craniotomy and secured to the skull with dental cement. Ear bars were removed, and the ferret was placed under the microscope for imaging.

### Mouse surgery
In vivo two-photon imaging was performed 3–6 weeks after viral injection. All mice were imaged at < 12 weeks of age, before the development of high frequency hearing loss (*Ison et al., 2007*). General anesthesia was induced with an intraperitoneal injection of ketamine (Vetalar; 100 mg/kg) and medetomidine (Domitor; 0.14 mg/kg), and was maintained with hourly subcutaneous injections of both agents (50 mg/kg/h Vetalar and 0.07 mg/kg/h Domitor). Body temperature was maintained at 37–38°C with a heating pad. The mouse was moved to a stereotaxic frame (Model 900LS, David Kopf Instruments), the scalp was incised, the temporal muscle retracted, and a craniotomy of ~2.5 mm diameter was performed over the injection sites in the right auditory cortex. The exposed area was covered with a glass coverslip, which was secured to the skull with cyanoacrylate adhesive (UltraGel; Pattex, DE). A steel holding post was attached to the left side of the skull using dental cement (UniFast Trad, GC Dental Products Corporation), ear bars were removed, and the mouse was placed under the microscope in the stereotaxic frame.

## Two-photon imaging

Imaging of calcium transients was performed using a B-Scope two-photon microscope (Thorlabs, Inc, UK) controlled by ScanImage 4.1 software (http://scanimage.org). Excitation light was emitted by a Mai-Tai eHP laser (SpectraPhysics, UK; 70 fs pulse width, 80 MHz repetition rate) tuned to 930 nm. The beam was directed into a Conoptics modulator (laser power, as measured under the objective, was 15–30 mW) and scanned through an 8 kHz resonant scanner in the x-plane and a galvanometric scanning mirror in the y-plane. The resonant scanner was used in bidirectional mode, at a resolution of 512 × 512 pixels, allowing us to acquire frames at a rate of ~ 30 Hz for our most common zoom. A 16X/0.80W LWD immersion objective (Nikon, UK) was used, and emitted photons were guided through a 525/50 filter onto GaAsP photomultipliers (Hamamatsu Photonics, Japan). Neuronal fields were between 200 × 200 μm and 300 × 300 μm in size. Neuronal activity was imaged in ferrets at 176 ± 26.83 μm (median ± s.d.) below the cortical surface, corresponding to layer 2 (*Dahmen et al., 2008*). In mice, imaging was performed at 216 ± 34.21 μm below the pial surface, corresponding to layer 2/3 in this species (*Anderson et al., 2009*).

## Sound presentation

Pure tones were generated via Matlab (MathWorks, Inc, USA), and an RZ6 multiprocessor (Tucker-Davis Technologies, USA) was used to synchronize the sound presentation with the microscope scanning.

Sound stimuli were presented binaurally to the animal via a customized closed acoustic delivery system comprised of either two Tucker-Davis Technologies EC1 electrostatic speakers (11 mice and three ferrets) or two Panasonic RPHV297 earphones (five ferrets). Speakers were coupled to a 12-cm-long silicone tube leading into the ear canal. The output response of the speakers was measured using a Brüel and Kjær calibration system with a GRAS 40DP microphone coupled to the end of the silicone tube. An inverse filter was applied to the speaker output to produce a flat spectral response (±3 dB) over the tone frequency range. Sound intensity was calibrated with an Iso-Tech TES-1356-G sound level calibrator.

The microscope and experimental animals were enclosed within a sound- and light-attenuating box. The ambient room noise was < 42 dB SPL inside this box, and primarily consisted of energy at < 200 Hz. The resonant scanner generated a constant acoustical tone of 8 kHz during imaging that was < 30 dB SPL near the animal's head.

For each imaging field in the ferret experiments, 10–20 repetitions of pure tones were presented at frequencies with 0.25 octave spacing across the ferret hearing range, and levels of 30–90 dB SPL, with 20 dB spacing (50 to 125 ms onset and offset cosine ramps). Frequency/level combinations were presented in pseudorandom order. Details of stimulus parameters for each animal are given in *Table 1*. For mice, ten repetitions of 100 ms duration (5 ms onset and offset cosine ramps) pure tones were presented at 18 frequencies (1.9–50 kHz, 0.6 octave spacing) and four levels (40, 60, 80, and 100 dB SPL), at a rate of 0.65 Hz.

**Table 1.** Details of stimuli presented to ferrets.

| Ferret | Stimulus duration (ms) | Repetitions | Interstimulus interval (s) | Frequency range (Hz) |
|---|---|---|---|---|
| 1 | 100 | 10 | 0.75 | 1259–41687 |
| 2 | 500 | 12 | 1.50 | 1047–28840 |
| 3 | 100 | 10 | 1.33 | 1259–41687 |
| 4 | 100 | 10 | 1.33 | 1259–41687 |
| 5 | 500 | 12 | 1.50 | 346–28840 |
| 6 | 500 | 20 | 1.50 | 346–28840 |
| 7 | 500 | 12 | 1.50 | 1047–28840 |
| 8 | 500 | 12 | 1.50 | 1047–28840 |

## Histology

At the end of each imaging session, experimental animals were overdosed (mice: 100 mg/kg ketamine and 0.14 mg/kg medetomidine, i.p.; ferrets: Euthatal, 1 ml pentobarbital sodium, i.p.) and perfused transcardially, first with 0.01 M phosphate-buffered saline (PBS) and heparin (20 units/ml), and then with 4% paraformaldehyde in PBS. Brains were removed and placed in 4% paraformaldehyde for two hours, after which they were stored in PBS with 0.01% sodium azide. Ferret brains were then cryoprotected in a 1:3 solution of sucrose and PBS for 24 hr. Sagittal brain sections (50 μm thickness) were obtained using a freezing sliding microtome (Leitz Wetzlar). Sections were sliced parallel to the orientation of the in vivo cranial window to facilitate the reconstruction of imaged neurons. Sections were washed three times in PBS, after which they were mounted onto microscope slides using Vectashield mounting medium (Vector Laboratories Ltd., USA). A coverslip was placed on the slide for imaging and sealed with clear nail polish.

## Confocal imaging

Brain sections were imaged using an inverted Olympus FV3000 six laser line spectral confocal microscope fitted with high sensitivity gallium arsenide phosphide (GaAsP detectors) and a 4x, 0.16 NA UplanSApo objective. The confocal pinhole was set to one airy unit to optimize optical sectioning with emission collection. Images were collected in resonant scanning mode at 512 × 512 pixels (pixel size 6.21 μm) and 16x averaging. Tile scans were stitched using the Olympus FluoView software.

## Identifying the location of GCaMP6 injection sites

A tracing of the ectosylvian and pseudosylvian sulci in ferret seven was used as a template of auditory cortex, and imaging fields from all eight ferrets were transferred onto this template. In ferrets 1, 5, 6 and 7, 1 mm x 1 mm images of the brain surface were acquired in vivo at the beginning of the two-photon imaging session, across the entire craniotomy. These images were then tiled together to reconstruct the surface of auditory cortex and locate the imaged fields. The tiled auditory cortex was then aligned with boundaries of the sulci in the auditory cortical template (Supplementary Figure 6a). For ferrets 2, 3, 4 and 8, GCaMP6 injection sites viewed on confocal images of the ferret ectosylvian gyrus (above) were aligned with the two-photon imaging fields acquired in the same ferret to precisely determine the location of each imaged field on the gyrus. The confocal images were then aligned to the sulci in the template map of the ectosylvian gyrus (*Figure 4—figure supplement 1*), using custom Matlab scripts. These procedures allowed us to plot the coordinates of neurons imaged in different animals onto a common map of auditory cortex.

## Data analysis for two-photon imaging experiments

### Isolating the responses of single neurons from imaging fields

We analysed the responses of 3604 neurons imaged in 32 imaging fields across eight ferrets, and 1962 neurons imaged in 42 imaging fields across 11 mice.

Videos of the imaging field during tone presentation were imported into Suite2p software (https://github.com/MouseLand/suite2p; *Pachitariu and Stringer, 2018*), which automatically performs mechanical drift correction, cell detection, and neuronal and neuropil trace extraction, and spike deconvolution. Parameters for the estimation of calcium transient templates were set separately for GCaMP6m and GCaMP6f, as the rise and decay times differ for these two indicators. These parameters were optimized using our own dataset. The most relevant parameter was the half decay time of the indicator (0.7 for GCaMP6f and 1.25 for GCaMP6m.

We manually inspected all regions of interest automatically detected by the Suite2p built-in classifier, in order to confirm if they were individual healthy neurons. This assessment was based on the frame-averaged image (e.g. the candidate neuron had a clear ring of florescence and was dark in the center), the alignment of the region of interest (e.g. the region did not expand beyond the cell membrane or include other neurons), and the activity of the calcium trace (e.g. no evidence of 'bursting' that characterizes cell death). For each confirmed neuron, time series of the neuronal calcium trace ($\Delta F/F_0$), neuropil calcium trace, and deconvolved spike probability were exported to Matlab for further analysis. Deconvolving the calcium trace allowed us to control for differences in the dynamics of the two calcium sensors used (GCaMP6m and GCaMP6f), and deconvolution

parameters (e.g. helf-decay constant) were adapted for each indicator, as described by *Pachitariu et al., 2018*.

## Identifying frequency sensitive neurons

Neuronal responses to sound presentation were quantified within a time window starting at stimulus onset and lasting twice as long as the stimulus. The 'evoked activity' was defined as the average of the inferred spike probability trace within the response window. A two-way ANOVA, with tone frequency and sound level as predictors, was used to determine if the evoked activity was significantly modulated by sound frequency or intensity ($\alpha$ = 0.05). Neurons showing a significant main effect of frequency or frequency/level interaction were defined as 'frequency sensitive', and only these neurons were included in further analyses.

## Calculating best frequency and classifying the frequency response

FRA plots were constructed for a given neuron from the trial-averaged evoked activity in response to tones presented at each frequency/level combination. For visualization, FRA plots were smoothed using a two dimensional three-point Gaussian kernel.

Frequency-response profiles of neurons were calculated by averaging the FRA across all sound levels. The frequency eliciting the highest response in the frequency-response profile was defined as the BF, as in previous studies (*Guo et al., 2012*; *Panniello et al., 2018*; *Barnstedt et al., 2015*). The FRA bandwidth, expressed in octaves, was defined as the continuous range of frequencies around BF that elicited a response greater than 50% of its maximal (i.e. BF) response.

Neurons were automatically categorized into three classes based on their frequency-response profiles. A neuron was defined as 'single-peaked' if its frequency-response profile contained only one continuous region above threshold, where threshold was defined as 75% of the maximal response across all frequencies. 'Double-peaked' neurons were those with two distinct regions above threshold in their frequency-response profile. For these FRAs, the frequency eliciting the strongest response was defined as $BF_d$, while the frequency eliciting the highest response in the other response region was defined as peak 2. A neuron was defined as 'complex' if it showed more than two discontinuous regions above threshold in the frequency-response profile. We also visually inspected FRAs (both smoothed and unsmoothed) of neurons and manually classified them as single-peaked, double-peaked or complex. The classification of 83% of neurons was the same under the automated frequency profile or manual FRA classification procedures. For the remaining 17% of neurons, the final classification was based on visual inspection of the FRA and frequency-response profile together.

## Trial-to-trial reliability

The trial-to trial reliability of responses at BF was estimated for each neuron as the Fano Factor (FF). That is, the variance in the evoked response to the BF across trials, divided by the average evoked activity at BF.

## Quantification of local BF variability

To assess the variability of frequency tuning within each imaging field, we computed the difference (in octaves) between each BF and the average BF of every other neuron in the imaging field, an approach similar to a recent study (*Tischbirek et al., 2019*). This analysis was performed independently on neurons from each of the three FRA classes, as well as on the three classes combined (i.e. all frequency-sensitive neurons). In this analysis, we included only imaging fields containing at least three single-peaked, three double-peaked and three complex neurons.

To compare the degree of local BF variability between mice and ferrets, it was necessary to account for differences in their hearing range and the length of their tonotopic gradient in A1. The mouse A1 is ~ 1 mm in length along the tonotopic gradient (*Guo et al., 2012*; *Stiebler et al., 1997*) and the hearing range of C57BL/6J mice at 10 weeks is 2–60 kHz (*Heffner and Heffner, 2007*; *Ison et al., 2007*), giving an A1 tonotopic map slope of 4.9 octaves/mm. In ferrets, the tonotopic gradient is ~ 3.5 mm long (*Kelly et al., 1986a*; *Bizley et al., 2005*) and the hearing range is ~ 36 Hz – 44 kHz (*Kelly et al., 1986b*), so the tonotopic gradient in A1 has a slope of 2.9 octaves/mm. For each imaging field, we first found the average BF across all the relevant neurons within the field (i.e.

all neurons of a given FRA class). The expected BF range was then calculated around this average BF, given the tonotopic slopes above and the size of the imaging field. The percentage of neurons having a BF within the expected range was used as a metric of variability of local frequency tuning within the field for that neuronal class.

### Signal and noise correlations

In keeping with previous studies (*Rothschild et al., 2010*; *Panniello et al., 2018*), signal and noise correlations were computed between the evoked responses of pairs of frequency-sensitive neurons recorded simultaneously. Briefly, noise correlations were estimated by first normalizing each neuron's response to sounds presented on individual trials by its signal response to that frequency/level combination. This normalization was obtained by subtracting, from each single trial response, the average response to all tones presented at that frequency/level combination. The noise correlation was then calculated across the normalized trial-by-trial responses of two cells.

Signal correlations were calculated as the correlation between the trial-averaged responses of any two neurons, minus the noise correlation computed for the neuronal pair.

### Analysing the effects of neuropil contamination

Neuronal calcium traces extracted from two-photon imaging acquisitions can be contaminated by the fluorescent signal coming from the pixels immediately surrounding each soma. For this reason, it is common practice to subtract from the neuronal trace what is known as the neuropil signal (*Chen et al., 2013*).

To test these effects, we calculated the BF of each neuron with and without neuropil subtraction, and the results are presented in *Figure 4—figure supplement 4*. The results were examined separately for the three FRA classes of neurons, and only for neurons that were frequency-sensitive (two-way ANOVA, described above) both with and without neuropil correction. The presence of the neuropil signal substantially changed the BF of some neurons (*Figure 4—figure supplement 4A*). Adding the neuropil signal caused a smaller change in the BF of single-peaked neurons ($BF_s$; 0.48 ± 0.05 octaves; mean ± SEM) than the BF of double-peaked neurons ($BF_d$; 0.76 ± 0.07 octaves; t-test: t = 3.36, p = $8.4 \times 10^{-4}$), peak 2 of double-peaked neurons (1.42 ± 0.09 octaves; t = 9.97, p = $2.3 \times 10^{-21}$), or the BF of complex neurons ($BF_c$; 1.08 ± 0.11 octaves; t = 5.94, p = $6.2 \times 10^{-9}$). There was a trend for the local variance in BF within an imaging field to be higher for double-peaked and complex neurons when the neuropil was subtracted, but these trends were not statistically significant (*Figure 4—figure supplement 4B*). When the BFs of 'neuropil contaminated' neurons were mapped onto the common template of A1, tonotopic organization was observed for single-peaked, double-peaked and complex cells (*Figure 4—figure supplement 4C*). Unlike in the neuropil-corrected signals (*Figure 4A3, B3 and C3*), the second frequency peak (peak 2) of double-peaked neurons also showed tonotopic organization when the neurons contained neuropil contamination (*Figure 4—figure supplement 4C3*, D3).

## In vivo electrophysiology

### Data acquisition

The animal preparation and anesthesia protocol was identical to the in vivo two-photon calcium imaging procedures described above. Recordings were carried out in the left auditory cortex. An Ag/AgCl external reference wire was inserted between the dura and the skull from the edge of craniotomy. After durotomy, the brain surface was covered with a solution of 1.25% agarose in 0.9% NaCl, and silicon oil was applied to the craniotomy regularly throughout recording.

A Neuropixels Phase 3a probe (*Jun et al., 2017*) was inserted orthogonally to the brain surface through the entire depth of auditory cortex. Data were acquired at a 30 kHz sampling rate using SpikeGLX software (https://github.com/billkarsh/SpikeGLX; *Karsh and Janelia Research Campus, 2017*) and custom Matlab scripts.

### Sound presentation

Electrophysiological recordings were made in a custom-built anechoic chamber. Stimuli were presented binaurally via Panasonic RP-HV094E-K earphone drivers, coupled to otoscope speculae inserted into each ear canal, and driven by a System 3 RP2.1 multiprocessor and headphone

amplifier (Tucker-Davis Technologies). The speculae were sealed in place with Otoform KC (Dreve Otoplastik GmbH). Speaker calibration was performed as described above for imaging experiments. Pure tones (0.5–40 kHz, 0.45 octave spacing, 110 ms duration, 5 ms cosine onset and offset ramps) were presented at five intensity levels (40–80 dB SPL). Each frequency/intensity combination was presented for 20 repetitions, in pseudorandomized order, at a rate 1.37 Hz.

### Spike sorting

The recorded signal was processed offline by first digitally highpass filtering at 150 Hz. Common average referencing was performed to remove noise across electrode channels (*Ludwig et al., 2009*). Spiking activity was then detected and clustered using Kilosort2 software (https://github.com/MouseLand/Kilosort2; *Pachitariu and Steinmetz, 2019*; *Pachitariu et al., 2016*). Responses from single neurons were manually curated using Phy (https://github.com/cortex-lab/phy; *Rossant et al., 2019*), if they had stereotypical spike shapes with low variance, and their autocorrelation spike histogram showed a clear refractory period. Spikes from a given cluster were often measurable on 4–6 neighboring electrode channels, facilitating the isolation of single units. Only well isolated single unit were included in subsequent analyses.

### Inverse current source density analysis

In order to directly compare the properties of neurons recorded with Neuropixels probes to those measured with two-photon calcium imaging, we aimed to isolate single units from cortical layers 2/3 in our electrophysiological recordings. To achieve this, we identified the boundary between layer 1 and 2 based on δ-source inverse Current Source Density analysis (iCSD) (*Pettersen et al., 2006*) in each recording penetration, and analyzed all single units down to a depth of 400 μm from this boundary.

The Local Field Potential (LFP) signal was isolated from the signal on each recording channel by bandpass filtering from 2 to 300 Hz, and notch filtering at 50 Hz to remove potential electrical noise. The evoked LFP trace was defined as time-series of the LFP signal starting 50 ms before tone presentation, and ending 50 ms after sound offset. Each LFP channel was impedance-normalized by subtracting the root mean squared power of the first 50 ms of the trial-averaged trace from the entire trace. This trace was averaged across repeated presentations and intensity levels for each tone frequency, and the frequency eliciting the strongest evoked LFP deflection was determined to be the best frequency for the channel. The iCSD of the recording was then computed from the LFP trace in response to tones presented at BF (±0.3 octaves), for all channels aligned vertically along the electrode using a published Matlab package (*Olsen, 2020*). The layer 1 to 2 boundary is characterized as a switch in polarity of the LFP (*Christianson et al., 2011*), and a sharp transition from a current source (layer 1) to a current sink (layers 2/3) in the iCSD (*Szymanski et al., 2009*; *Szymanski et al., 2011*; *Cooke et al., 2018*). For all A1 penetrations, there was a clear reversal in LFP polarity at the boundary between layers 1 and 2 (*Figure 2—figure supplement 2*), as previously described (*Christianson et al., 2011*; *Szymanski et al., 2011*).

### Calculating the best frequency of single units

Single units were identified on channels from the upper border of layers 2/3 (based on current source density analysis) to depths 400 μm below this border. The evoked spike rate for each neuron on each trial was calculated as the sum of spikes from tone onset to offset. The proportion of frequency-sensitive neurons (two-way ANOVA; $p<0.05$), BF, and FRA classifications of these evoked spike responses were calculated as described above for calcium imaging data.

## Acknowledgements

We are grateful to Prof Patrick Kanold for his advice and help in setting up our initial two-photon calcium imaging experiments in ferrets. Thanks to Dr Ben Willmore for assistance with electrophysiological experiments. Thanks also to the MICRON imaging facility (http://micronoxford.com, supported by Wellcome Strategic Awards 091911/B/10/Z and 107457/Z/15/Z) for technical advice and access to equipment. This work was supported by a BBSRC New Investigator Award (BB/M010929/1) and a DPAG Early Career Fellowship (University of Oxford) to KMMW, a Christopher Welch Scholarship

(Oxford University Press) to AZI, a Newton-Abraham Scholarship (University of Oxford) to MP, and a Wellcome Principal Research Fellowship to AJK (WT076508AIA, WT108369/Z/2015/Z).

## Additional information

### Competing interests
Andrew J King: Senior editor, *eLife*. The other authors declare that no competing interests exist.

### Funding

| Funder | Grant reference number | Author |
|---|---|---|
| Biotechnology and Biological Sciences Research Council | BB/M010929/1 | Kerry MM Walker |
| University of Oxford | DPAG Early Career Fellowship | Kerry MM Walker |
| Wellcome | WT076508AIA | Andrew J King |
| Wellcome | WT108369/Z/2015/Z | Andrew J King |
| University of Oxford | Christopher Welch Scholarship | Aleksandar Z Ivanov |
| University of Oxford | Newton-Abraham Scholarship | Mariangela Panniello |

The funders had no role in study design, data collection and interpretation, or the decision to submit the work for publication.

### Author contributions
Quentin Gaucher, Data curation, Software, Formal analysis, Investigation, Methodology, Writing - review and editing; Mariangela Panniello, Conceptualization, Data curation, Software, Formal analysis, Investigation, Methodology, Writing - original draft, Writing - review and editing; Aleksandar Z Ivanov, Data curation, Software, Formal analysis, Investigation, Methodology; Johannes C Dahmen, Data curation, Methodology, Writing - review and editing; Andrew J King, Conceptualization, Supervision, Funding acquisition, Writing - review and editing; Kerry MM Walker, Conceptualization, Data curation, Software, Formal analysis, Supervision, Funding acquisition, Investigation, Methodology, Writing - original draft, Project administration, Writing - review and editing

### Author ORCIDs
Johannes C Dahmen ⬤ http://orcid.org/0000-0001-9889-8303
Andrew J King ⬤ https://orcid.org/0000-0001-5180-7179
Kerry MM Walker ⬤ https://orcid.org/0000-0002-1043-5302

### Ethics
Animal experimentation: The animal procedures were approved by the University of Oxford Committee on Animal Care and Ethical Review and were carried out under license from the UK Home Office, in accordance with the Animals (Scientific Procedures) Act 1986 and in line with the 3Rs. Project licence PPL 30/3181 and PIL l23DD2122.

### Decision letter and Author response
Decision letter https://doi.org/10.7554/eLife.53462.sa1
Author response https://doi.org/10.7554/eLife.53462.sa2

## Additional files

### Supplementary files

• Supplementary file 1. Supplementary statistics for t-tests and ANOVAs described in the main figures.

• Transparent reporting form

### Data availability

We have provided our data and Matlab scripts for generating our figures on Dryad: https://doi.org/10.5061/dryad.9ghx3ffd9.

The following dataset was generated:

| Author(s) | Year | Dataset title | Dataset URL | Database and Identifier |
|---|---|---|---|---|
| Gaucher Q, Panniello M, Ivanov AZ, Dahmen JC, King AJ, Walker KMM | 2019 | Complexity of frequency receptive fields predicts tonotopic variability across species | https://doi.org/10.5061/dryad.9ghx3ffd9 | Dryad Digital Repository, 10.5061/dryad.9ghx3ffd9 |

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
