## [Decision Letter]

**Acceptance summary:**

This study elegantly compares spatial organisation of frequency tuning in auditory cortex across two very different species, mice and ferrets, which had never been done. Beyond showing marked similarities between the two species, this work highlights the complexity of frequency receptive as a major factor contributing in two species to the apparent local disorganisation of the frequency map. This strongly reinforces the view, that irrespective of the species, auditory cortex not only implements a map of frequency but also builds elaborate representations of sounds likely useful for capturing the acoustic diversity of real-world auditory stimuli.

**Decision letter after peer review:**

Thank you for submitting your article "Complexity of frequency receptive fields predicts tonotopic variability across species" for consideration by *eLife*. Your article has been reviewed by three peer reviewers, including Brice Bathellier as the Reviewing Editor and Reviewer #1, and the evaluation has been overseen by Richard Ivry as the Senior Editor. The following individual involved in review of your submission has agreed to reveal their identity: Daniel B Polley (Reviewer #3).

The reviewers have discussed the reviews with one another and the Reviewing Editor has drafted this decision to help you prepare a revised submission.

Summary:

Using both calcium imaging and Neuropixel electrophysiology , this study shows that in ferrets, the tonotopic map is locally heterogenous, and that double and multi-peak tuning curves contribute more to tonotopic map heterogeneity than single peak cells. The study also shows that multi-peak cells have also a best frequency less attuned to the surrounding BFs in mice, but that double peak cells have the same degree of heterogeneity as single peak cells.

This paper provides a new important dataset, with the first two-photon imaging observations in AC in a carnivorous species, complemented with Neuropixel data. It also provides interesting analyses suggesting that the source of tonotopic map heterogeneity is the increasing complexity of frequency receptive fields in the auditory cortex, particularly in ferrets. The comparison of species (ferret vs mouse) is a strength of the study and the comparison of methods (single unit electrophysiology versus 2p calcium imaging) as well. The most interesting and surprising conclusion is that the ferret and mouse have comparable degrees of local heterogeneity in best frequency tuning. The main discovery of the paper – that the orderly arrangement of tonotopy in L2/3 predominantly arises from cells with well-defined tuning to a single of frequencies – has already been discovered at least twice before (Guo et al., 2012; Romero et al., 2019), but the study here confirms and extends this observation to a different species and with a slightly different analysis.

Yet there are several non-trivial – but solvable – problems with the data analysis that deserve careful consideration because they could affect many of the main conclusions. In addition, the reviewer noted several issues with the accuracy, clarity, scholarship and general writing style.

Essential revisions:

1) The point that double and multipeak cells are strong contributors to the local imprecision of tonotopic is convincing for ferrets, but less in mice. The discrepancy may be due to less precise frequency tuning in mice (broader tuning curves), which may lower the fraction of multipeak cells (as peaks would be less well defined).

It will be useful to compare tuning width in both species, especially for single-peak cells. Indeed, if we follow the authors' idea, that the more imprecision there is in the BF definition, the more likely it is to be far from the mean local BF, then tuning width would be an important factor.

For example, the authors could provide a graph for both species in which some generalized bandwidth (say the fraction of the frequency range above half max response) is plotted against distance to mean frequency. All three categories could be displayed in the graph so that one could evaluate if the broadly tuned cells in mice tend to be the one cells that are not following tonotopy.

2) Authors use both GCaMP6m and GCaMP6f. It is well known that GCaMP6f underreports spiking. How does this affect the observed tuning diversity? One would expect that the tuning diversity is similar to that of GCaMP3.

3) There is a need to improve the statistical criteria for determining whether a neuron was frequency-sensitive. The authors use a 2-way ANOVA with frequency and level as factors and label a neuron as frequency-sensitive only if there is a main effect for frequency or a significant freq x level interaction term. There is a concern about Type-II error because a neuron either with very narrow tuning (e.g. responds to only 1-2 frequencies) or a neuron with very broad tuning and low-threshold would quite possibly not be included in the analysis. Because only 20% of cells were found to be frequency sensitive, this deserves consideration, particularly because the most important factor in this paper is to contrast cells with very well-defined tuning versus cells with broad/complex tuning. Looking at Figure 1C, for example, it seems clear that they are excluding many cells with frequency sensitivity. It makes much more sense to include cells in the analysis based on whether they are responsive to the tone (e.g. a paired t-test of peak amplitude in the pre-stimulus baseline vs post-stimulus period across trials) and then break them down according to their frequency/level preference in the next stage of analysis.

4) There are some serious issues with the way BF variance is calculated.

4a) There is a problem with using the mean BF within the ROI instead of the median. The Tischbirek and Romero studies used the median, not the mean to avoid the high frequency artifact that comes from performing a linear operation (mean) on log2 data (BF). The authors should take the mean of the log2 BF values (to avoid high frequency BF artifact) or just take the median, which has the added benefit of being less sensitive to outliers.

4b) The bigger issue is that if they calculated variance as a function of number of neurons within the set I suspect you would find a main effect for the number of neurons (more variance with smaller number of constituent neurons). If so, this is a confound as the number of units co-varies with single vs double vs complex FRA types. The way to solve this would be to bootstrap the measurement using a fixed number of neurons (e.g., always three).

5) Measurement of sound-evoked activity. Is this measure averaged across all levels at BF or measured at the conjunction of BF and BL? If the former, it would also be confounded by the number intensities that elicited a response (i.e., would be impacted by thresholds or for O-type neurons).

6) The CSD is troublesome. Apart from a discontinuity at L1 (which might just be the pial surface) all is seen is one distributed current sink across layers. One should see an early current sink (i.e., negative value) in layer 4 surrounded by a current source in L2/3 and L5/6 that then flips in polarity with time after stimulus onset. This pattern has been seen in dozens of papers, at least when the CSD is measured as the second spatial derivative (it was not clear if this is what they did). Generally, either the Materials and methods are missing a description of filtering/smoothing methods that are typically used for CSD analysis or else the description was correct and the problem is that the analysis is incomplete. The authors should refer to papers by Lakatos and Schroeder, Metherate and colleagues or Guo and Polley for a more detailed description of data processing for CSD analysis.

7) The signal correlation measurement is not convincing. If two units don't share the same set of frequency/level combinations in their FRAs their signal correlations could be lower. Alternatively, if the sound-evoked activity rates are systematically lower, the signal correlations could also be lower. This muddies the interpretation of what the signal correlation finding means. Instead, the authors might consider identifying the frequency-intensity combinations that fall inside the FRA for every neuron. Then they can take a pair of neurons and calculate the fraction of shared freq/intensity combinations that they share as a function of distance. With this approach, they are at least measuring receptive field similarity independent of systematic differences in activity rate.

Also the signal and noise correlations may be better placed in the supplementary figures.

8) This is more of a presentation problem, but generally speaking the presentation of the mouse data feels rushed and superficial. No information is given on where the images are being acquired, nor are we given any indication of the image or signal quality as was shown for the ferret. It seems entirely possible to me that they are not recording from mouse A1. Some anatomical data that shows the GCaMP expression relative to known landmarks for A1 in the mouse (as described in Romero et al., 2019) along the lines of what is carefully shown for the ferret in Figure 1—figure supplements 1 and 2 would help alleviate this concern somewhat.

The bottom line is that the cross-species comparison is one of the most interesting points of the paper. Counting all of the supplements, we get around 10 figures on the ferret data and then just a single combined figure on the mouse and the cross-species comparison.

[Editors' note: further revisions were suggested prior to acceptance, as described below.]

Thank you for resubmitting your work entitled "Complexity of frequency receptive fields predicts tonotopic variability across species" for further consideration by *eLife*. Your revised article has been evaluated by Richard Ivry as the Senior Editor, Brice Bathellier as the Reviewing Editor, and two reviewers.

The manuscript has been improved and all reviewers are satisfied with the clarifications provided. However there is one remaining issue that needs to be addressed before acceptance, as outlined below:

The revised manuscript benefits from many improvements. In their response to major issue #3 (pasted below), the authors missed the main point and instead chose to focus on the choice of the test. The main point was that it seemed like they were undercounting the number of neurons that changed their activity rates when tones were presented. Figure 1C bears this out in that plenty of cells below their significance threshold have visible tuning, so type II error (too conservative test) can be an issue. The authors should show to which extent relaxing the significance threshold / type of test (which always has some arbitrariness) to include more cells changes (or not) their conclusions. This could be done by including all cells, or maybe the cells that pass a simple t-test for baseline against response but pooling all sounds together, or by using any other evaluation of global signal-to-noise ratio and varying the threshold.

Previous comment:

"3) There is a need to improve the statistical criteria for determining whether a neuron was frequency-sensitive. The authors use a 2-way ANOVA with frequency and level as factors and label a neuron as frequency-sensitive only if there is a main effect for frequency or a significant freq x level interaction term. There is a concern about Type-II error because a neuron either with very narrow tuning (e.g. responds to only 1-2 frequencies) or a neuron with very broad tuning and low-threshold would quite possibly not be included in the analysis. Because only 20% of cells were found to be frequency sensitive, this deserves consideration, particularly because the most important factor in this paper is to contrast cells with very well-defined tuning versus cells with broad/complex tuning. Looking at Figure 1C, for example, it seems clear that they are excluding many cells with frequency sensitivity. It makes much more sense to include cells in the analysis based on whether they are responsive to the tone (e.g. a paired t-test of peak amplitude in the pre-stimulus baseline vs post-stimulus period across trials) and then break them down according to their frequency/level preference in the next stage of analysis.”

---

## [Author Response]

Essential revisions:1) The point that double and multipeak cells are strong contributors to the local imprecision of tonotopic is convincing for ferrets, but less in mice. The discrepancy may be due to less precise frequency tuning in mice (broader tuning curves), which may lower the fraction of multipeak cells (as peaks would be less well defined).It will be useful to compare tuning width in both species, especially for single-peak cells. Indeed, if we follow the authors' idea, that the more imprecision there is in the BF definition, the more likely it is to be far from the mean local BF, then tuning width would be an important factor.For example, the authors could provide a graph for both species in which some generalized bandwidth (say the fraction of the frequency range above half max response) is plotted against distance to mean frequency. All three categories could be displayed in the graph so that one could evaluate if the broadly tuned cells in mice tend to be the one cells that are not following tonotopy.

We thank the reviewers for this interesting suggestion. We implemented the reviewers’ recommendation to calculate tuning bandwidth in both species. We defined the frequency tuning bandwidth for each neuron as the continuous frequency span around BF that evoked a response >50% of the neuron’s maximal response. This definition of tuning width has been added to the Materials and methods section:

“The FRA bandwidth, expressed in octaves, was defined as the continuous range of frequencies around BF that elicited a response greater than 50% of the neuron’s maximal (i.e. BF) response.”

As the reviewers predicted, the tuning width of single-peaked neurons was significantly broader in mice than ferrets (t-test; t = 3.66, p = 2.7 x 10^-4^). There were no species differences, however, in the tuning widths of double-peaked or complex neurons. As the reviewers suggested, it is possible that broader frequency tuning in mice may have led to an increased proportion of single-peaked neurons in this species, because the dominant peak in the profile may have occupied a larger extent of the maximum bandwidth.

A graph comparing tuning width in ferrets and mice has been added Figure 5—figure supplement 2A, and reported in the Results:

“Neurons with single-peaked FRAs showed broader frequency tuning in mice (0.91 ± 0.03 octaves) compared to ferrets (0.74 ± 0.03 octaves; t-test: t = 3.66, p = 2.7 x 10^-4^). […] However, there were no significant species differences in tuning bandwidth for double-peaked (t = 0.76, p = 0.44) or complex neurons (t = 0.72, p = 0.47; Figure 5—figure supplement 2A).”

As the reviewers suggested, we also now provide a scatter plot of BF variability (BF distance from median BF in an imaging field) as a function of tuning bandwidth for single-peaked neurons (Figure 5—figure supplement 2B, C). This plot shows that there was no significant correlation between tuning bandwidth and the local BF variability in ferrets (r=0.02; p=0.726; Pearson’s correlation). In mice, there was a weakly significant correlation between these two variables (r=-0.11; p=0.030), but the direction of this correlation was opposite to the reviewers’ prediction. In both ferrets and mice, tuning bandwidth was not significantly correlated with BF variability in either double-peaked or complex neurons (data not shown; p > 0.05). Therefore, there is no evidence that broader frequency tuning accounts for larger variance in local BF. These results are reported in the Results:

“Furthermore, larger bandwidths of single-peaked neurons were not associated with larger deviations in local BF in ferrets (Pearson’s correlation: r = 0.020, p = 0.73; Figure 5—figure supplement 2B), and were weakly correlated with smaller BF deviations in mice (r = -0.11, p = 0.030; Figure 5—figure supplement 2C). […] Therefore, the sharpness of frequency tuning does not explain local BF variability.”

2) Authors use both GCaMP6m and GCaMP6f. It is well known that GCaMP6f underreports spiking. How does this affect the observed tuning diversity? One would expect that the tuning diversity is similar to that of GCaMP3.

The reviewers are correct in pointing out that GCaMP6f is a less sensitive reporter of action potentials than GCaMP6m. Indeed, in our GCaMP6f experiments in ferrets, we found that only 237 of 2192 (10.81%) imaged neurons were frequency sensitive, while 456 of 1412 imaged neurons were frequency sensitive when using GCaMP6m (32.29%). To examine if differences in sensitivity between the two reporters led to different frequency tuning, we repeated the main analyses, this time treating data from GCaMP6m and GCaMP6f ferrets separately. As expected, all our main results were replicated in these subsets of data, demonstrating that they are robust to the sensitivity of the GCaMP6 indicator. This justifies our choice to pool the data from both reporters in our main manuscript. The results of these analysis are included in the new Figure 2—figure supplement 2. Specifically, we compare the differences in local BF variability, the activity at BF, and the Fano Factor across single-peaked, double-peaked and complex neurons.

We describe these analyses in the Results:

“As expected, when data from GCaMP6m and GCaMP6f injections were analysed separately, we found a higher percentage of frequency-sensitive neurons in the GCaMP6m (32.29%), compared to the GCaMP6f dataset (10.81%). Importantly, both indicators showed similar effects of FRA class on local BF variance, response strength and response reliability (Figure 2—figure supplement 2), so our main findings are consistent across the two indicators, and data are pooled across all ferrets for our remaining analyses.”

3) There is a need to improve the statistical criteria for determining whether a neuron was frequency-sensitive. The authors use a 2-way ANOVA with frequency and level as factors and label a neuron as frequency-sensitive only if there is a main effect for frequency or a significant freq x level interaction term. There is a concern about Type-II error because a neuron either with very narrow tuning (e.g. responds to only 1-2 frequencies) or a neuron with very broad tuning and low-threshold would quite possibly not be included in the analysis. Because only 20% of cells were found to be frequency sensitive, this deserves consideration, particularly because the most important factor in this paper is to contrast cells with very well-defined tuning versus cells with broad/complex tuning. Looking at Figure 1C, for example, it seems clear that they are excluding many cells with frequency sensitivity. It makes much more sense to include cells in the analysis based on whether they are responsive to the tone (e.g. a paired t-test of peak amplitude in the pre-stimulus baseline vs post-stimulus period across trials) and then break them down according to their frequency/level preference in the next stage of analysis.

We agree with the reviewers that the method used to define frequency-selective ROIs is a crucial step in our study, and indeed any study of stimulus sensitivity. We had therefore already examined the use of several different inclusion criteria in our analyses. The reassuring and most important result of these analyses was that our main findings (i.e. local variability in BF across different FRA classes, differences in response strength at BF, etc) were robust across all initial “responsivity” inclusion metrics that we have tested.

To address the reviewers’ suggestion specifically, here we compare our test of frequency sensitivity (i.e. the 2-way ANOVA) to the t-test of tone responsiveness that the reviewers suggested. For each neuron, the responses (pooled across all tones) during a pre-stimulus time period (300 ms up to stimulus onset) were compared to the response during the post-stimulus periods (300 ms from stimulus onset) using a paired t-test (alpha=0.05). Neurons passing this t-test were defined as “responsive”. A comparison of the results of this t-test to our original 2-way ANOVA (alpha = 0.05) is presented in Author response image 1. The Venn diagrams show the number of significant ROIs detected by the 2-way ANOVA and t-test for each species.

In ferrets (Author response image 1A), while some ROIs passed both tests (32%), overall the ANOVA selected more significant ROIs (ANOVA: 70%; t-test:62%), in contrast to the reviewers’ expectations. For both tests, we were able to detect single-peaked, double-peaked and complex FRAs (Author response image 1B), although the proportion of complex cells was higher with a ROI selection based on the ANOVA. Finally, the effects of FRA class on local BF variability for the neurons selected with the t-test (Author response image 1C) was equivalent to those described for ANOVA-selected ROIs (Figure 2C in the main manuscript).

In mice, a larger proportion of ROIs passed both tests (57%; Author response image 1D) compared to ferrets. Perhaps unsurprisingly given the overlap between the two datasets, the proportions of single-peaked, double-peaked and complex neurons were very similar when calculated using the ANOVA or t-test (Author response image 1E). Finally, the effects of FRA class on local BF variability observed when using the t-test (Author response image 1F) were the same as those reported using the ANOVA (Figure 5D in the main manuscript).

These comparisons show that using a t-test or an ANOVA to define tone-responsive neurons does not change the main results of our study. Contrary to the reviewer’s concern, a higher number of significant ROIs in ferrets were included by our ANOVA than the alternative t-test. Furthermore, as we want to examine frequency tuning in the present paper, the ANOVA test of frequency-sensitivity is more appropriate than a t-test, which only asks if neurons respond to the stimuli (possibly in a frequency-independent manner). We hope that in light of these new comparative analyses, the Reviewing Editor and reviewers will agree that it is sensible to continue to use our 2-way ANOVA as the ROI inclusion test to define frequency-sensitive neurons.

**Author response image 1. sa2fig1:** Comparison of 2-way ANOVA and paired t-test as ROI inclusion tests. (A) Venn diagram showing the overlap in the number of ferret ROIs passing the ANOVA (pink) and the t-test (green). (B) Proportions of single-peaked, double-peaked and complex FRAs within the population of neurons identified using either the ANOVA (left) or t-test (right). (C) Cumulative probability plots of the difference (in octaves) between each neuron’s BF and the median BF of all neurons of the same FRA class in the same imaging field. Distributions for different FRA classes are plotted separately as colored lines, as in Figure 2C. Data are for neurons identified as responsive by the t-test. D, E, F. Mouse data plotted as in A, B, C (respectively) for ferret data above.

4) There are some serious issues with the way BF variance is calculated.4a) There is a problem with using the mean BF within the ROI instead of the median. The Tischbirek and Romero studies used the median, not the mean to avoid the high frequency artifact that comes from performing a linear operation (mean) on log2 data (BF). The authors should take the mean of the log2 BF values (to avoid high frequency BF artifact) or just take the median, which has the added benefit of being less sensitive to outliers.

We thank the reviewers for this excellent suggestion. We had originally used the mean in accordance with previous papers (including the Tischbirek study that the reviewers cite, incidentally). But the reviewers are correct that the median is a better measure. We have now calculated local BF variance within an imaging field using the median BF instead of the mean BF. Accordingly, we have replaced Figure 2C, Figure 2—figure supplement 2A and D, Figure 4—figure supplement 3B1, 2, 3, 4, Figure 5D, E, F with new plots using the median BF calculation. This has naturally changed the relevant statistical values in our manuscript somewhat, and these have been updated throughout. They are too numerous to list here, but are highlighted in yellow in the manuscript. Our results and interpretations all remain robust using this new metric. We find the same effects of FRA classes on BF local and global variability.

4b) The bigger issue is that if they calculated variance as a function of number of neurons within the set I suspect you would find a main effect for the number of neurons (more variance with smaller number of constituent neurons). If so, this is a confound as the number of units co-varies with single vs double vs complex FRA types. The way to solve this would be to bootstrap the measurement using a fixed number of neurons (e.g., always three).

The reviewers express a reasonable concern about the effect that number of neurons can have on estimates of BF variance. To address this concern, we have carried out the analyses suggested and show the results in the new Figure 5—figure supplement 3. This includes scatter plots of the average BF variability (BF distance to median BF) of single-peaked neurons within an imaging field, as a function of the number of neurons within that field, calculated for the ferret (**A**) and mouse (**D**) datasets. There was no significant correlation between BF variability and the number of neurons in the field, either in ferrets (r = -0.30, p = 0.20; **A**) or mice (r = 0.33, p = 0.25; **D**). In addition, we carried out the bootstrapping analysis suggested by the reviewers. For each imaging field, we estimated the BF variance (around the mean BF) using only 3 neurons for every possible combination of 3 neurons. We used the mean BF instead of the median BF for this analysis because the median of 3 BFs would lead to a variance of 0 for at least 1 ROI out of 3. Like the raw BF variance reported in the manuscript, the average bootstrapped BF variance was larger for double-peaked neurons than single-peaked neurons, and larger in complex neurons compared to double-peaked neurons in both species (ferrets: **B**; mice: **E**). In addition, the cumulative histograms of the bootstrapped BF variances also replicate the results described in our manuscript for both species (ferrets: **C**; mice: **F**).

Therefore, the analyses suggested by the reviewers confirmed our original results. We believe that, given the consistency of the results obtained with the two methods, it is more appropriate to use the real number of neurons in the main manuscript in Figure 2C, Figure 2—figure supplement 2A and D, Figure 4—figure supplement 3B1, 2, 3, 4 and Figure 5D. We have added this figure to our manuscript as Figure 5—figure supplement 3, and have added the accompanying text to the Results:

“To ensure that the smaller number of multi-peaked neurons in mice did not bias our estimates of BF variance, we repeated this analysis using the same number of neurons in each imaging field. […] The same patterns of BF variance were found across single-peaked, double-peaked and complex cells in these control analyses (Figure 5—figure supplement 3B, C, E and F).”

5) Measurement of sound-evoked activity. Is this measure averaged across all levels at BF or measured at the conjunction of BF and BL? If the former, it would also be confounded by the number intensities that elicited a response (i.e., would be impacted by thresholds or for O-type neurons).

The sound-evoked activity and Fano Factor were measured at the conjunction of best frequency and best level. We apologize that this wasn’t clear and have modified the text to clarify.

“We found that the average deconvolved calcium response at the best frequency and level combination was stronger in single-peaked neurons compared to either double-peaked neurons (t-test: t = 2.22, p = 0.027) or neurons with complex FRAs (t = 4.02, p = 6.9 x 10^-5^). […] The Fano Factor calculated at the best frequency and level did not significantly differ between neurons in the three FRA classes (single- and double-peaked: t = 0.60, p = 0.55; single-peaked and complex: t = 0.638, p = 0.52), indicating that responses at BF were equally reliable for neurons with single- and multi-peaked FRAs (Figure 2E).”

6) The CSD is troublesome. Apart from a discontinuity at L1 (which might just be the pial surface) all is seen is one distributed current sink across layers. One should see an early current sink (i.e., negative value) in layer 4 surrounded by a current source in L2/3 and L5/6 that then flips in polarity with time after stimulus onset. This pattern has been seen in dozens of papers, at least when the CSD is measured as the second spatial derivative (it was not clear if this is what they did). Generally, either the Materials and methods are missing a description of filtering/smoothing methods that are typically used for CSD analysis or else the description was correct and the problem is that the analysis is incomplete. The authors should refer to papers by Lakatos and Schroeder, Metherate and colleagues or Guo and Polley for a more detailed description of data processing for CSD analysis.

We apologize for the omission of details in our Materials and methods, which led to this confusion. We did not use the classic CSD analysis method (Nicholson and Freeman, 1975) of taking the second spatial derivative of the local field potential (LFP) that the reviewers describe. Instead, we used the δ-source inverse current source density (iCSD) method previously described by Pettersen et al. (Pettersen et al., 2006). A detailed mathematical description of the iCSD method is provided in that paper, especially in their Figure 13. We have modified our description of our CSD and re-labelled it as “iCSD” in our manuscript, to make these methods clearer.

The iCSD method was chosen over the standard CSD method because it gives a much clearer demarcation of the boundary between cortical layer 1 and 2. This boundary is characterized by a clear switch in the polarity of the LFP (Christianson et al., 2011). With the standard CSD method, this boundary is not as clear, e.g. see Figure 1A (monkey data; Lakatos et al., 2007), Figure 3 col 3 (gerbil data; Schaefer et al., 2017) and Figure 2C (mouse data; Guo et al., 2017). The iCSD method provides a better representation of this boundary, e.g. Figure 4 bottom row (rat data; Szymanski et al., 2009), Figure 1C (rat data; Szymanski et al., 2011) and Figure 3B (mouse data; Cooke et al., 2018). The temporal and spatial profile of our auditory cortical responses are typical for the iCSD analysis described in these 3 previous papers (Szymanski et al., 2009, Szymanski et al., 2011, Cooke et al., 2018).

Our iCSD profile shown in Figure 2—figure supplement 3 resembles that previously reported in rat barrel cortex (Pettersen et al., 2006) and in both rat (Szymanski et al., 2009) and mouse (Cooke et al., 2018) auditory cortex, with a sharp switch in polarity at the transition between the current source in layer 1 and the current sink in layers 2/3, and a larger current sink in the principal thalamorecipient layer 4.

References:

-Guo W, Clause AR, Barth-Maron A, Polley DB (2017) A Corticothalamic Circuit for Dynamic Switching between Feature Detection and Discrimination. Neuron 95:180-194.e5 Available at: https://linkinghub.elsevier.com/retrieve/pii/S0896627317304579.

-Lakatos P, Chen C-M, O’Connell MN, Mills A, Schroeder CE (2007) Neuronal Oscillations and Multisensory Interaction in Primary Auditory Cortex. Neuron 53:279–292 Available at: https://linkinghub.elsevier.com/retrieve/pii/S0896627306009962.

-Nicholson C, Freeman JA (1975) Theory of current source-density analysis and determination of conductivity tensor for anuran cerebellum. J Neurophysiol 38:356–368 Available at: https://www.physiology.org/doi/10.1152/jn.1975.38.2.356.

-Schaefer MK, Kössl M, Hechavarría JC (2017) Laminar differences in response to simple and spectro-temporally complex sounds in the primary auditory cortex of ketamine-anesthetized gerbils Malmierca MS, ed. PLoS One 12:e0182514 Available at: http://dx.plos.org/10.1371/journal.pone.0182514.

“Inverse Current Source Density analysis

In order to directly compare the properties of neurons recorded with Neuropixels probes to those measured with two-photon calcium imaging, we aimed to isolate single units from cortical layers 2/3 in our electrophysiological recordings. […] For all A1 penetrations, there was a clear reversal in LFP polarity at the boundary between layers 1 and 2 (Figure 2—figure supplement 2), as previously described (Christianson, Sahini and Linden, 2011; Szymanski et al., 201).”

7) The signal correlation measurement is not convincing. If two units don't share the same set of frequency/level combinations in their FRAs their signal correlations could be lower. Alternatively, if the sound-evoked activity rates are systematically lower, the signal correlations could also be lower. This muddies the interpretation of what the signal correlation finding means. Instead, the authors might consider identifying the frequency-intensity combinations that fall inside the FRA for every neuron. Then they can take a pair of neurons and calculate the fraction of shared freq/intensity combinations that they share as a function of distance. With this approach, they are at least measuring receptive field similarity independent of systematic differences in activity rate.Also the signal and noise correlations may be better placed in the supplementary figures.

The reviewers express concern that signal correlations will be lower for neurons if their evoked activity is lower. Algebraically, this is not the case. Correlations are independent of the magnitude of signals, so the overall firing rates of neurons do not affect signal correlations. Scaling the values in one or both sets of responses from a given neuron does not change its resulting correlation with a second neuron, nor does adding an absolute constant value. We demonstrate this empirically in Author response image 2. We multiplied all the responses of one neuron in each pair by 2, thereby doubling its sound-evoked activity rate. The resulting signal correlations (y-axis) are exactly equal to the original signal correlations (x-axis).

**Author response image 2. sa2fig2:** Pairwise signal correlations for all pairs of simultaneously recorded ferret neurons. The signal correlation (SC) was calculated as in our original manuscript (x-axis) or after doubling the response strength of one neuron in each pair (y-axis). This scaling of the response strength has no effect on the resulting signal correlation. Signal correlations of individual pairs of neurons are plotted as blue dots, and these all fall on the line of equality (red).

While simply scaling the response of neurons does not change their signal correlations, the reviewers might instead be concerned that the responses of neurons with lower response rates become more difficult to detect or estimate, leading to sampling or estimation biases that could in turn affect signal correlations. To examine this possibility, we plotted our pairwise signal correlations as a function of the combined response strength of both neurons in the pair, to test the reviewers’ suggestion that less responsive neurons may have lower signal correlations. The results are shown in Author response image 3. Contrary to the reviewers’ suggestion, signal correlations in the ferret data were largest for pairs of neurons with weaker signals. This effect seems to result from the small number of highly responsive neurons in each case, which tend to have poor signal correlations. Indeed, if we limit our analysis to neuron pairs with summed responses below 1 (which includes the majority of our neuron pairs), there is no systematic relationship between these two plotted values. In mice, only single-peaked neurons showed this trend, and there was no correlation between the response strength and signal correlations for double-peaked and complex neurons. In summary, these graphs show no evidence of poorer signal correlations among weakly-responding neurons.

**Author response image 3. sa2fig3:** Pairwise signal correlations as a function of the summed response strength of paired neurons. The signal correlation was calculated as in our original manuscript (y-axis), and is plotted as a function of the response strength of the pair of neurons (x-axis). The response strength was calculated as the two neurons’ summed extracted spikes, each averaged across all stimulus conditions (x-axis). Individual neuron pairs are shown as blue dots and the regression line is overlaid. The results of the regression of signal correlation on response strength is given below each panel title. Results are plotted separately for ferrets (A, B, C) and mice (D, E, F), and for single-peaked (A, D), double-peaked (B, E) and complex (C, F) cells.

Finally, we implemented the reviewers’ suggested alternative measure of tuning similarity. For each neuron’s FRA, we calculated which frequency/level combinations elicited a response that exceeded half the maximum response. We then measured the overlap in these “response-eliciting” stimuli between pairs of simultaneously imaged neurons. The results are shown in Author response image 4. The results support those reported in our manuscript using signal correlations. In both cases, there is evidence of decreasing signal correlations with increased distance for single-peaked and double-peaked neurons, but not for complex neurons. Furthermore, single-peaked and double-peaked neurons have more similar FRAs within an imaging field than complex neurons using either the FRA overlap or signal correlation metric. This demonstrates that these findings are not due to some problem with signal correlations.

**Author response image 4. sa2fig4:** FRA overlap between pairs of simultaneously recorded neurons, plotted as a function of their distance along the tonotopic gradient. Y-axis shows the number of stimuli eliciting a response >50% maximum for both neurons in a given pair. Each blue dot represents a pair of neurons. The red line is the fitted linear regression, and statistics for the fit are provided below the titles above each plot. Results are plotted separately for ferrets (A, B, C) and mice (D, E, F), and for single-peaked (A, D), double-peaked (B, E) and complex (C, F) cells.

8) This is more of a presentation problem, but generally speaking the presentation of the mouse data feels rushed and superficial. No information is given on where the images are being acquired, nor are we given any indication of the image or signal quality as was shown for the ferret. It seems entirely possible to me that they are not recording from mouse A1. Some anatomical data that shows the GCaMP expression relative to known landmarks for A1 in the mouse (as described in Romero et al., 2019) along the lines of what is carefully shown for the ferret in Figure 1—figure supplements 1 and 2 would help alleviate this concern somewhat.The bottom line is that the cross-species comparison is one of the most interesting points of the paper. Counting all of the supplements, we get around 10 figures on the ferret data and then just a single combined figure on the mouse and the cross-species comparison.

The reviewers are correct in noting that we present a much more detailed analysis of the ferret data compared to the mouse data. This is because the ferret data are all new, while the mouse data have been previously published in Panniello et al., 2018, as reported in the subsection “Mouse surgery”. The Reader can find the details that the reviewer enquires about in this 2018 publication, including the anatomical and functional methods used to confirm imaging location in the primary auditory cortex.

We have expanded our Materials and methods to summarize these validations of imaging locations in the mouse.

“To confirm that we were targeting mouse A1 with our viral vector injections, the coordinates used matched those of fluorescent retrobead injections that retrogradely labelled neurons in the ventral division of the medial geniculate body (Panniello et al., 2018). […] Nevertheless, it is not possible to say with the same degree of confidence as for the ferret data that all mouse imaging reported in the present study was performed in A1.”

We agree with the reviewers that providing more details about the mouse data in the present manuscript would improve its clarity and strengthen our conclusions. Therefore, we have added two supplementary figures (Figure 5—figure supplements 1 and 2) to report additional new analyses of the mouse data for comparison with the ferret data.

Figure 5—figure supplement 1A shows single trial calcium traces from 1 neuron along with the corresponding FRA, as shown for a ferret neuron in Figure 1B. Figure 5—figure supplement 1B shows FRAs of 9 further neurons in one mouse to illustrate single, double and complex frequency tuning in the mouse dataset (as shown for ferrets in Figure 2A). This figure also shows for the mouse data the response strengths (Figure 5—figure supplement 1C; as shown for ferrets in Figure 2D) and Fano Factor calculations (Figure 5—figure supplement 1D; as shown for ferrets in Figure 2E).

We describe these results in the new text:

“Single-peaked and double-peaked neurons in mice responded more strongly at BF than complex neurons (single- and double-peaked: t = -0.73, p = 0.46; single-peaked and complex: t = 3.20, p = 0.0010; double-peaked and complex: t = 3.43, p = 1.9 x 10^-4^ ; Figure 5—figure supplement 1C), and trial-to-trial reliability in complex neurons was better than that of single-peaked neurons (single- and double-peaked: t = -1.60, p = 0.11; single-peaked and complex: t = 2.08, p = 0.037; Figure 5—figure supplement 1D).”

Figure 5—figure supplement 2 and accompanying new text are discussed above in response to point 1.

[Editors' note: further revisions were suggested prior to acceptance, as described below.]

The manuscript has been improved and all reviewers are satisfied with the clarifications provided. However there is one remaining issue that needs to be addressed before acceptance, as outlined below:The revised manuscript benefits from many improvements. In their response to major issue #3 (pasted below), the authors missed the main point and instead chose to focus on the choice of the test. The main point was that it seemed like they were undercounting the number of neurons that changed their activity rates when tones were presented. Figure 1C bears this out in that plenty of cells below their significance threshold have visible tuning, so type II error (too conservative test) can be an issue. The authors should show to which extent relaxing the significance threshold / type of test (which always has some arbitrariness) to include more cells changes (or not) their conclusions. This could be done by including all cells, or maybe the cells that pass a simple t-test for baseline against response but pooling all sounds together, or by using any other evaluation of global signal-to-noise ratio and varying the threshold.Previous comment:"3) There is a need to improve the statistical criteria for determining whether a neuron was frequency-sensitive. The authors use a 2-way ANOVA with frequency and level as factors and label a neuron as frequency-sensitive only if there is a main effect for frequency or a significant freq x level interaction term. There is a concern about Type-II error because a neuron either with very narrow tuning (e.g. responds to only 1-2 frequencies) or a neuron with very broad tuning and low-threshold would quite possibly not be included in the analysis. Because only 20% of cells were found to be frequency sensitive, this deserves consideration, particularly because the most important factor in this paper is to contrast cells with very well-defined tuning versus cells with broad/complex tuning. Looking at Figure 1C, for example, it seems clear that they are excluding many cells with frequency sensitivity. It makes much more sense to include cells in the analysis based on whether they are responsive to the tone (e.g. a paired t-test of peak amplitude in the pre-stimulus baseline vs post-stimulus period across trials) and then break them down according to their frequency/level preference in the next stage of analysis.”

We thank the reviewers for clarifying their original point. As described in previous reports from many labs, a substantial proportion of neurons in primary auditory cortex are unselective or unresponsive to pure tones. For this reason, we believe that using a statistical test to segregate frequency-sensitive neurons is necessary when attempting to investigate the spatial organization of frequency tuning in A1. Such an approach has been followed in many previous studies (Rothschild et al., 2010; Tischbirek et al., 2019). Nevertheless, we acknowledge the concern about type II error. To address this concern, we now re-analyse both our mouse and our ferret datasets assigning a best frequency to all imaged neurons, regardless of their frequency sensitivity. Our main findings (i.e. local variability in BF across different FRA classes, global variability of BF along the tonotopic axis, species differences in the percentage of neurons in each FRA class) remain when all neurons imaged are included in the analysis. Therefore, these findings are robust across the three inclusion criteria now examined: (1) our original two-way ANOVA testing for frequency sensitivity; (2) a t-test for tone responsivity suggested by the reviewers, and presented in our previous response letter; and (3) including all imaged neurons. The main difference in the results of these 3 approaches is that including all neurons results in a higher proportion of neurons with “complex” FRAs, in both species, which one would expect if noisy data are classified.

We report the results of including all imaged neurons in a new supplementary figure (Figure 5—figure supplement 4), and have added the accompanying text in our Results.

“As many neurons in sensory cortex can be unresponsive or poorly tuned to pure tone frequency, it is common practice to map only neurons that are significantly modulated by sound frequency (Rothschild, Nelken and Mizrahi, 2010; Tischbirek et al., 2019). […] Furthermore, more scatter in the spatial distribution of BF was observed with increased FRA complexity at both local (Figure 5—figure supplement 4B and C) and global (Figure 5—figure supplement 4D-G) levels of tonotopic organization, as we found for frequency-sensitive neurons (Figure 2C; Figure 4C; Figure 5D).”